# Nanoscale architecture of a VAP-A-OSBP tethering complex at membrane contact sites

Eugenio de la Mora[1,2,5], Manuela Dezi [1,2,5✉], Aurélie Di Cicco[1,2], Joëlle Bigay [3], Romain Gautier [3], John Manzi [1,2], Joël Polidori[3], Daniel Castaño-Díez[4], Bruno Mesmin [3], Bruno Antonny [3✉] & Daniel Lévy [1,2✉]

Membrane contact sites (MCS) are subcellular regions where two organelles appose their membranes to exchange small molecules, including lipids. Structural information on how proteins form MCS is scarce. We designed an in vitro MCS with two membranes and a pair of tethering proteins suitable for cryo-tomography analysis. It includes VAP-A, an ER trans-membrane protein interacting with a myriad of cytosolic proteins, and oxysterol-binding protein (OSBP), a lipid transfer protein that transports cholesterol from the ER to the trans Golgi network. We show that VAP-A is a highly flexible protein, allowing formation of MCS of variable intermembrane distance. The tethering part of OSBP contains a central, dimeric, and helical T-shape region. We propose that the molecular flexibility of VAP-A enables the recruitment of partners of different sizes within MCS of adjustable thickness, whereas the T geometry of the OSBP dimer facilitates the movement of the two lipid-transfer domains between membranes.

[1] Laboratoire Physico Chimie Curie, Institut Curie, PSL Research University, CNRS UMR168, Paris, France. [2] Sorbonne Université, Paris, France. [3] CNRS UMR 7275, Université Côte d'Azur, Institut de Pharmacologie Moléculaire et Cellulaire, Valbonne, France. [4] BioEM Lab, C-CINA, Biozentrum, University of Basel, Basel, Switzerland. [5] These authors contributed equally: Eugenio de la Mora, Manuela Dezi. ✉email: manuela.dezi@curie.fr; bruno.antonny@ipmc.cnrs.fr; daniel.levy@curie.fr

Membrane contact sites (MCS) are subcellular regions where two organelles associate to carry out non-vesicular communication[1–6]. MCS involve almost every organelle, are present in all tissues, play important roles in lipid exchange, calcium signaling, organelle fission, inheritance, and autophagy, and are implicated in metabolic diseases.

In MCS, the two facing membranes are closely apposed, typically 15–30 nm apart, over distances up to micrometers. Consequently, MCS are highly confined spaces but their molecular organization is poorly understood, notably with regards to protein stoichiometry, density, orientation, and dynamics. Proteins involved in membrane tethering display a wide variety of organizations: single or multiple polypeptide chains, complexes of two proteins, each of them associated with a different organelle, or multimeric assemblies such as the ERMES complex between the ER and mitochondria[1].

To date, no complete structure of tethers involved in MCS formation has been solved. In general, structural information is limited to protein domains such as those involved in lipid transfer or organelle targeting[7–13]. Because these domains are often connected by disordered linkers, this leaves numerous possibilities for how the full-length proteins orient and move between the two facing membranes. Moreover, all structures have been determined in the absence of membranes, resulting in an incomplete view of MCS.

Thanks to advances in cryo-electron microscopy (cryo-EM) and in situ cryotomography (cryo-ET), first images of MCS at medium resolution have been obtained. Synaptotagmins and their yeast orthologs tricalbins are rod-shape 15–20 nm long structures that bridge the ER and the PM[14–16].

Here, we address the general question of the formation of MCS at a molecular scale by studying a model MCS formed by VAP-A and OSBP. VAP-A and its homolog VAP-B are transmembrane ER proteins that interact with ≈100 cytosolic partners. They do so through a common mechanism: the cytosolic Major Sperm Protein (MSP) domain of VAP-A/B recognizes FFAT (two phenylalanine in an acidic track) or FFAT-like motifs in partner proteins[17–21]. The large spectrum of VAP-A interactants and their involvement in many MCS make the structure of VAP-A at MCS an important open question.

OSBP and OSBP-related proteins (ORPs) constitute a large family of lipid transfer proteins (LTPs). Several ORPs transport-specific lipids in a directional manner owing to the counter exchange and hydrolysis of the phosphoinositide PI4P. OSBP that contains an FFAT motif and that binds VAP-A drives cholesterol/PI4P exchange at ER-Golgi MCS[22,23] and is the target of several anticancer and antiviral compounds[24,25], pointing to its key role in cellular homeostasis.

In this work, we designed an in vitro system adapted for cryo-EM and cryo-ET analysis of MCS formed by VAP-A and either OSBP or a shorter construct, N-PH-FFAT, containing OSBP tethering determinants. By sub-tomogram averaging, we obtained 3D models of VAP-A and N-PH-FFAT between facing membranes, which reveal the organization of membrane tethering.

## Results

We studied the structure of minimal MCS formed by two purified proteins, VAP-A and OSBP. The domain organization of these proteins is schematized in Fig. 1a.

VAP-A is inserted in the ER membrane using a single C-terminal transmembrane helix (TM) and exposes its N-terminal globular domain of the MSP family to the cytoplasm. The MSP, which recognizes proteins that contain FFAT or FFAT-like motifs, is separated from the TM by a predicted coiled-coil region (CC)[13,19]. Together with the TM, the CC promotes the dimerization of VAP-1/B proteins[26].

OSBP contains five functional regions: an N-terminal intrinsically disordered region, a PH domain, a putative central CC domain, an FFAT motif, and a C-terminal OSBP-related domain (ORD), which exchanges cholesterol for PI4P[23,27]. OSBP bridges ER and TGN via its FFAT motif, which recognizes VAP-A, and via its PH domain, which recognizes Arf1-GTP and/or PI4P present in the Golgi membrane[23]. A simplified N-PH-FFAT construct of OSBP, lacking the ORD, recapitulates the tethering function of OSBP[23,28]. N-PH-FFAT and OSBP were purified as reported[23].

**Purification/biochemical and functional characterization of full-length VAP-A.** In this study, we used full-length OSBP or N-PH-FFAT to reconstitute MCS and analyze their architecture by cryo-EM and cryo-ET. To better mimic the cellular conditions, we used full-length VAP-A with its TM inserted into the liposome bilayer.

We expressed full-length VAP-A in *Escherichia coli* and purified it in a micellar form using the mild detergent n-dodecyl-β-D-maltoside (DDM) (Fig. 1b). Size exclusion chromatography showed a single peak at MW ~ 420 kDa. As observed for membrane proteins, the apparent MW was higher than that expected for a VAP-A monomer (27.9 kDa) or dimer (54.8 kDa) owing to the presence of the DDM micelle. In sodium dodecyl sulphate–polyacrylamide gel electrophoresis (SDS-PAGE) gel and in the presence of reducing agents, VAP-A preparation showed two bands at ~55 kDa and ~25 kDa, whereas a single band was found when the samples were heated revealing dimers of VAP-A in the preparation (Supplementary Fig. 1a).

We reconstituted VAP-A in liposomes by adding to the VAP-A in DDM, a solubilized mixture of egg phosphatidylcholine (PC) and brain phosphatidylserine (PS) at a 95/5 mol/mol, a ratio similar to what found at the ER, followed by detergent removal (Fig. 1c). We chose a high lipid/protein ratio (LPR ≈ 1400 mol/mol) to ensure that proteins were present at low density in the membrane. Cryo-EM showed that the proteoliposomes were spherical, unilamellar, and displayed a diameter ranging from 40 to 200 nm. After flotation on a sucrose gradient, the proteoliposomes were recovered in a single low sucrose density band, suggesting a homogeneous population (Fig. 1c, lanes 6–11). No protein aggregates were found at the bottom of the gradient (lane 23 in Fig. 1c).

To test if VAP-A in proteoliposomes was able to functionally interact with OSBP (Fig. 1d), we used an assay that follows the transfer of PI4P from Golgi-like to ER-like liposomes in real time[23]. Previous biochemical reconstitutions and cellular observations indicated that interaction with membrane-bound VAP-A determines the lipid exchange activity of OSBP[23].

We incubated the VAP-A proteoliposomes with Golgi-like liposomes containing 4% PI(4)P and Rhodamine lipid (Rho-PE), and used the probe NBD-PH as a fluorescent reporter of the membrane distribution of PI(4)P. At the beginning of the experiment, the fluorescence of NBD-PH was quenched by Rhodamine lipid (Rho-PE), as both PIP4 and Rho-PE were present in the same liposomes. The addition of OSBP triggered a large increase in the fluorescence of NBD-PH due to the transfer of PI(4)P to the VAP-A proteoliposomes, which did not contain Rho-PE (Fig. 1d). PI4P transfer rate increased with the amount of OSBP and was inhibited by OSW1, a specific OSBP inhibitor.

These results show that full-length VAP-A purified in DDM is well folded, can be incorporated in lipid membranes, and is functional.

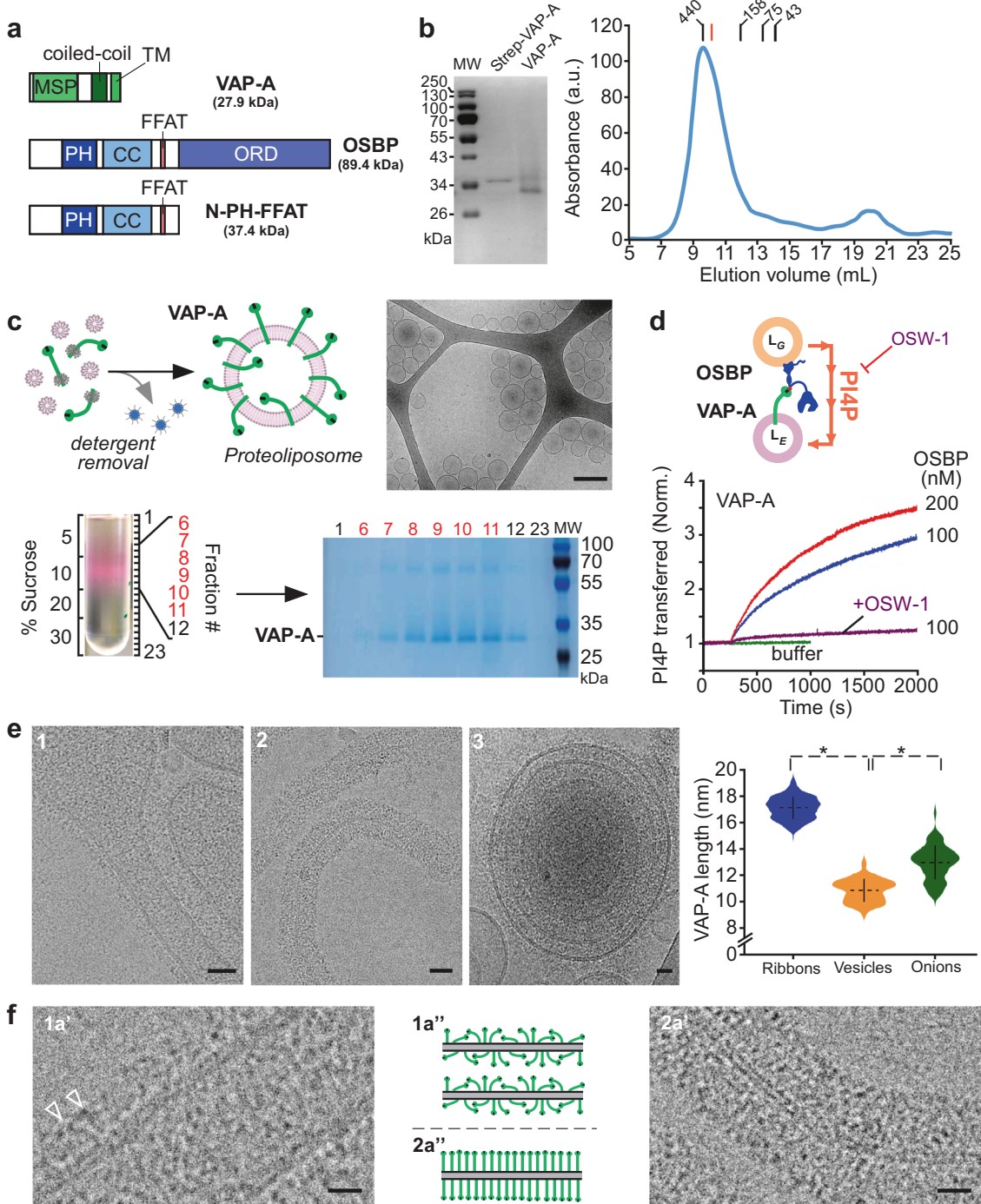

## VAP-A extends at increasing distances from the membrane with its concentration

We reconstituted VAP-A at LPRs ranging from 2800 to 50 mol/mol. We expected that increasing VAP-A density could favor protein–protein interaction in the membrane plane. We visualized the reconstituted proteoliposomes by cryo-EM. When the starting mixture was at LPR 70 mol/mol, i.e a high protein density, the reconstituted proteoliposomes were heterogeneous in shape and size. We observed (i) spherical vesicles, (ii) non-spherical vesicles displaying angular shapes or tubulation, (iii) fragments of open membranes resembling ribbons with proteins on both side of the bilayer, and (iv) multilayered vesicles (Fig. 1e, Supplementary Fig. 1b, f). When the concentration of VAP-A was reduced to LPR 350 mol/mol, the fraction of spherical vesicles increased at the expense of ribbons and deformed

vesicles (Supplementary Fig. 1c). At high LPR (700–1400 mol/mol), all proteoliposomes were spherical or slightly deformed (Supplementary Fig. 1d, e). Thus, vesicle morphology depended on the concentration of VAP-A in the membranes. High protein density shapes the membrane probably by crowding effect and/or lateral protein–protein interactions, as shown for other transmembrane proteins reconstituted at high density for electron crystallography and AFM[29,30].

Even though VAP-A has a small size for cryo-EM, electron densities of VAP-A extending out of the membrane of vesicles were clearly visible. We measured the length of the extramembrane domain of VAP-A in different types of vesicles after reconstitution at LPR 70 mol/mol using the well-resolved electron density of the external lipid leaflet. VAP-A extended $10 \pm 2$ nm

**Fig. 1 VAP-A extends from the membrane with increasing protein density. a** Domain organization of VAP-A, of OSBP, and of the N-PH-FFAT construct. **b** Purification of full-length VAP-A with its transmembrane domain after solubilization in n-dodecyl-β-D-maltoside (DDM). SDS-PAGE analysis of purified strepII-TEV-tag VAP-A before and after TEV proteolysis. Size exclusion chromatography of VAP-A. The red marker is BmrA, a 130 kDa membrane protein solubilized in DDM. **c** The scheme describes the principle of proteoliposome formation by detergent removal. Lipid/protein molar ratio (LPR) before detergent removal was adjusted to change the amount of VAP-A in proteoliposomes. Floatation of VAP-A proteoliposomes doped with a fluorescent lipid. SDS-PAGE analysis shows that VAP-A is incorporated in a single population of vesicles. Cryo-EM images of VAP-A proteoliposomes. Fractions from the sucrose gradient were not heated at 95 °C as done in 1B in the denaturation buffer explaining the presence of the band at 60 kDa corresponding to the non-dissociated dimer of VAP-A. Bar = 250 nm. **d** The scheme describes the principle of PI4P translocation between Golgi-lipid vesicles and VAP-A proteoliposomes as catalyzed by OSBP. Real-time measurement of PI(4)P transfer in the presence of VAP-A proteoliposomes at LPR 4000 mol/mol. NBD-PH (3 μM) was mixed with Golgi-like liposomes (250 μM lipids containing 4 mol % PI(4)P and 2 mol % rhodamine lipid (Rho-PE)) and VAP-A proteoliposomes (250 μM lipids, 0.17 nM Vap-A). At $t = 300$ sec, OSBP was added at 100 or 200 nM as indicated. OSW1 (1 μM) an inhibitor of OSBP was used to inhibit PI4P transfer. **e** Reconstitution of VAP-A at high protein density (LPR 70 mol/mol) leading to the formation of deformed vesicles (1), ribbons (2), onions (3). Violin plots showing the length distribution of the extramembrane region of VAP-A. The plot shows the complete distribution of values analyzed. The horizontal and vertical lines show the average value and the standard deviation, respectively. ∗ indicates $p < 0.01$ by unpaired $t$ test. 50 vesicles ($n = 78$ measurements), 45 ribbons ($n = 421$ measurements), and 6 onions ($n = 78$ measurements) were analyzed. Bar = 25 nm. **f** Close-up views of regions 1a′, 2a′ where electron densities of VAP-A are a visible and schematic representation of protein concentration and orientation in the membrane in deformed vesicles 1a″ 2a″ (tubular region of e1) and ribbons (e2). Some individual proteins are resolved (white arrowheads). The extramembrane region of VAP-A is visible and extends at increasing distances from the membrane. Bars = 10 nm (**e**, **f**). Figure 1e. Type of analysis, unpaired (independent), two-sided (two tails), $t$ test, valid for all plots. $p$ p value, CI confidential interval. Plot vesicles vs onions: $p = 3.3788e-23$, CI = [−2.4715, −1.7642], plot onions vs ribbons $p = 6.8058e-14$, CI = [−4.3753, −3.9233]. A representative result or micrograph is shown from at least three replicates.

($n = 284$ measurements, 50 vesicles) from the outer lipid leaflet of deformed vesicles. In some cases, individual proteins with an elongated shape were identified (Fig. 1f, 1a′–1a″, white arrows). At the protein tip, dark dots were followed by a region toward the membrane without a defined structure. Previous structural studies[13,26,31] and our model of VAP-A (see below Fig. 4) suggest that the dark distal densities are likely to correspond to the MSP domain. These dark dots were also found closer to the membrane suggesting that VAP-A and/or its MSP domains adopted various orientations with regard to the membrane plane. In the case of ribbons (Figs. 1f, 2a′–2a″), VAP-A appeared more compact and perpendicular to the lipid bilayer, extending up to $17 \pm 2$ nm ($n = 422$ measurements, 50 ribbons).

The multilayered vesicles were made of concentric membrane layers separated by protein densities, suggesting that they were held together by VAP-A homotethers (Fig. 1e3). This organization was reminiscent of that of *Ascaris suum* MSP dimers, which assembled antiparallel owing to 3D crystal packing[32]. VAP-B homotether existence has been suggested to explain stacked ER cisternae phenotype in Nir2/VAP-B-expressing cells[33]. The separation between facing membranes in multilayered vesicles was 22–28 nm, suggesting that VAP-A molecules extend 11–14 nm from the membrane.

In summary, VAP-A extends at variable distances from the membrane, adopting either a compact and tilted conformation at low surface density or an extended conformation at high surface density, suggesting a highly flexible molecule.

**Formation of MCS with VAP-A and OSBP**. We next studied how VAP-A binds partners and forms a MCS.

Because VAP-A and N-PH-FFAT are small, we suspected that the putative VAP-A/N-PH-FFAT complex of 130.6 kDa might be difficult to recognize within a contact zone where proteins are densely packed. We thus designed an in vitro system that could allow us to assign the membrane system to which each protein was bound. Specifically, we mixed two morphologically different membranes: VAP-A was incorporated in egg PC/brain PS liposomes, whereas N-PH-FFAT was bound to lipid tubes made of galactocerebroside (GalCer) containing PI(4)P (Fig. 2a). The lipid tubes were also doped with PC and PS to increase membrane fluidity[34]. They had a constant diameter of 27 nm and showed variable lengths of several hundreds of nanometers (Supplementary Fig. 2a). N-PH-FFAT or OSBP were bound at

high concentrations to the tubes as shown by the protein densities covering the external lipid leaflet and extending 5–6 nm from the membranes.

We incubated VAP-A proteoliposomes with PI4P tubes decorated with N-PH-FFAT and froze the mixture for cryo-EM observation. After 2 min incubation, we observed massive aggregation of vesicles and tubes suggesting fast tethering of the two membranes (Supplementary Fig. 2b, d). To reduce aggregation, we added sequentially both components directly onto the cryo-EM grid, before freezing within <30 seconds (Supplementary Fig. 2c, e). Contact areas were recognizable at low magnification with remodeled vesicles in close proximity of tubes (Supplementary Fig. 2e). This protocol resulted in an ice layer that was thin enough for cryo-EM or cryo-ET and was therefore selected for further analysis of MCS architecture and formation.

**The MCS intermembrane distance depends on the local concentration of VAP-A**. VAP-A is present along with the ER and concentrates in MCS where FFAT-containing proteins are present[23,35]. However, the number of VAP-A molecules engaged in MCS is not known. We analyzed how MCS assembled when the concentration of VAP-A in the membrane varied.

We reconstituted VAP-A at a low LPR ratio (70 mol/mol) to obtain a mix of spherical vesicles, deformed vesicles, and membrane ribbons.

After incubation with N-PH-FFAT tubes, different types of contact areas were observed depending on the morphology of the VAP-A-containing bilayer. We identified (i) angular-shaped proteoliposomes with high protein density on the surface and with membrane lying along the tubes, (ii) hemispherical vesicles with deformations restricted to a part of the vesicles, and (iii) spherical liposomes in tangential contact with the tubes (Fig. 2b, d). The separation between the facing membranes was not constant and depended on the class of contacting vesicles. Measurements from 2D images showed that the intermembrane distance between the tubes and the VAP-A-containing membranes increased from 15 nm ± 5 nm for the spherical vesicles to 20 ± 5 nm for the highly deformed vesicles (Fig. 2c). Larger distances up to 30 nm were measured in cryo-tomograms of VAP-A ribbons (Fig. 2d). This trend was similar to that observed for increasing concentrations of VAP-A in membranes, suggesting an effect of VAP-A density on membrane separation in MCS.

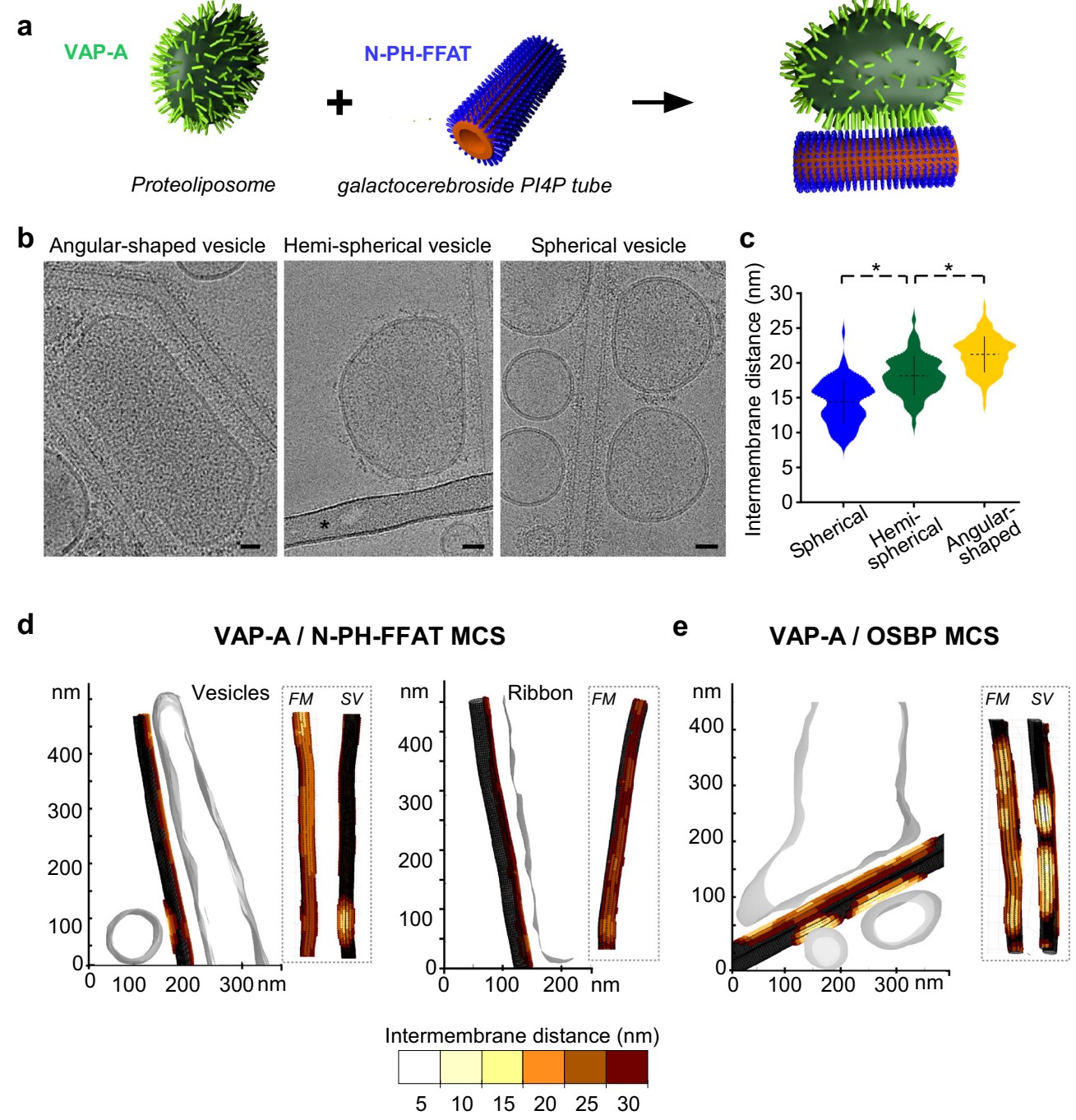

**Fig. 2 VAP-A density tunes the separation between facing membranes in reconstituted membrane contact sites (MCS). a** Scheme of the reconstitution of MCS between VAP-A proteoliposomes and N-PH-FFAT (or OSBP) bound to galactocerebroside tubes doped with PI4P. **b** Representative images of the different types of contacts made between VAP-A proteoliposomes (lipid/protein ratio LPR 70 mol/mol) and N-PH-FFAT bound to tubes. The bar labeled with * in the central panel is a carbon grid bar. Bars = 25 nm. **c** Violin plots of the distance between facing membranes in reconstituted MCS (LPR 70 mol/mol). The plots show the complete distribution of values analyzed. The horizontal and vertical lines show the average value and the standard deviation, respectively. ∗ indicates $p < 0.01$ by unpaired $t$ test. Intermembrane distance between tubes and spherical vesicles ($n = 90$), hemispherical vesicles ($n = 122$), and angular-shaped vesicles ($n = 128$) were analyzed. **d** 3D reconstruction of VAP-A/N-PH-FFAT and **e** VAP-A/OSBP MCS from cryo-tomograms. Distances between angular shape and spherical vesicles and tubes and ribbons and tubes are depicted in color codes on tubes. *FM* indicates the tube side in contact with a flattened region of large vesicles and *SV* the tube side in contact with small spherical vesicles. Figure 2c. Type of analysis, unpaired (independent), two-sided (two tails), $t$ test, valid for all plots. $p$ $p$ value, *CI* confidential interval. Plot spherical vs hemispherical vesicles: $p = 3.9588e{-}1$, CI [−4.5561, −2.9464]. Plot hemispherical vs angular-shaped: $p = 1.3549e{-}20$, CI = [−3.6377, −2.4378]. A representative micrograph or result is shown from at least three replicates.

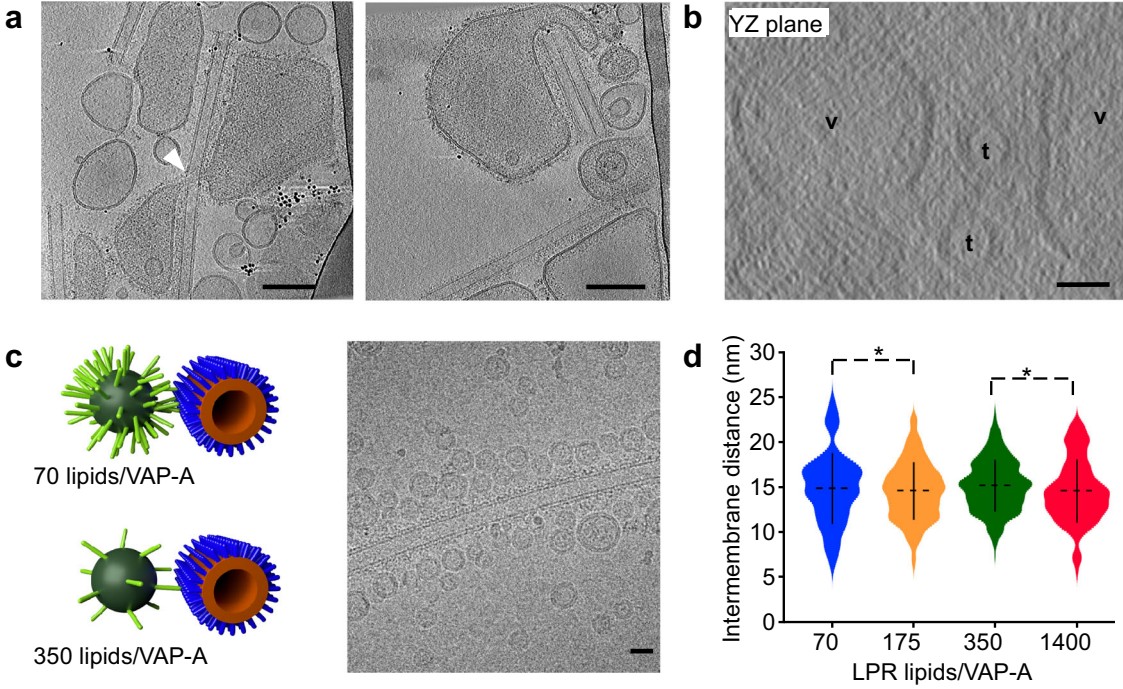

**Fig. 3 Membrane deformation upon MCS formation. a** Tomographic XY slice from two tilt series showing that VAP-A vesicles spread along the long axis of tubes, form tongue (left figure, white arrow) around the tubes, but in general avoid the high curvature short axis of tubes (right figure). Bars = 100 nm. **b** Tomographic YZ slice from a third tilt series showing the flat membrane of vesicles (v) facing tubes (t). Bars = 25 nm. **c** Small and non-deformable VAP-A proteoliposomes at varying LPR and bound to N-PH-FFAT tubes. Representative MCS with VAP-A vesicles of 30 nm diameter at 70 and 350 lipids/VAP-A mol/mol. Cryo-EM image of small liposomes of VAP-A at LPR350 lipids/VAP-A forming contact with N-PH-FFAT coated tubes. Bar = 25 nm. **d** Violin plots showing the distribution of distances between small vesicle of VAP-A at varying LPR and N-PH-FFAT tube. The plots show the complete distribution of values analyzed. The horizontal and vertical lines show the average value and the standard deviation, respectively. ∗ indicates $p < 0.05$ by unpaired $t$ test. Intermembrane distance between tubes and vesicles at LPR 70 ($n = 55$), LPR175, ($n = 240$), LPR350 ($n = 491$), and LPR1400 ($n = 152$) were analyzed. Figure 3d. Type of analysis, unpaired (independent), two-sided (two tails), $t$ test, valid for all plots. $p = p$ value, $CI = $ confidential interval. Plot LPR175 vs LPR350: $p = 0.0145$, $CI = [−1.0367, −0.1143]$. Plot LPR350 vs LPR1400: $p = 0.0375$, $CI = [0.0341, 1.1404]$. A representative result is shown from at least three replicates.

**MCS made with VAP-A can accommodate proteins of different sizes**. To assess whether the presence of the bulky ORD (42 kDa) of OSBP modified the MCS architecture, we compared MCS formed with either N-PH-FFAT or full-length OSBP.

We first reconstituted VAP-A at LPR 70 mol/mol to obtain proteoliposomes with various VAP-A densities. Then, they were mixed either with N-PH-FFAT or OSBP bound to PI4P Galcer tubes. The resulting MCSs were analyzed by cryo-ET. As shown by trypsin digestion, VAP-A was symmetrically oriented in proteoliposomes (Supplementary Fig. 1g) but only VAP-A pointing outward from the vesicles was engaged in MCS formation.

Figure 2d and Supplementary Movies 1, 2 show tomographic reconstruction of an MCS formed by VAP-A and N-PH-FFAT as determined by cryo-ET. The 3D reconstruction revealed a long contact region of ca. 500 nm between a large VAP-A vesicle and a tube, as well as a small contact region of <50 nm with a spherical vesicle (see also Fig. 3a, c and Supplementary Fig. 3 for other examples). As observed in 2D images, the separation between facing membranes was larger when a flattened VAP-A-containing vesicle was in contact with tubes, compared to a spherical VAP-A vesicle; the intermembrane distances were 20–25 nm and 15 nm, respectively. We also found VAP-A ribbons forming contact regions, which represent the membrane system with the highest VAP-A density, formed MCS with tubes with an intermembrane distance of 25–30 nm. Both flattened vesicles and ribbons showed an even separation along the whole MCS, suggesting homogenous distribution of tethers.

MCS made with OSBP showed differences with that observed with N-PH-FFAT. First, OSBP promoted smaller surfaces of contact between tubes and vesicles, even when large VAP-A vesicles were spread along tubes (Fig. 2e, supplementary Movie 3). Second, spherical VAP-A vesicles in which VAP-A is present at low density were involved in 48% of OSBP MCSs ($n = 89$) as compared with 30% in the case of the MCSs induced by N-PH-FFAT MCSs ($n = 226$). In contrast, no MCSs were found between OSBP and the VAP-A ribbons, i.e., the membrane in which VAP-A is present at the highest density. Third, the membrane separations in MCSs induced with OSBP were shorter than that observed with N-PH-FFAT. In flat contacting regions, between proteoliposomes and tubes, we measured intermembrane distances of 10–20 nm in the presence of OSBP, as compared with 15–25 nm in the presence of N-PH-FFAT.

All these results suggest that the ORD of OSBP imposes space between proteins, thereby preventing high packing and elongation of VAP-A proteins.

**Membrane remodeling during the formation of MCS**. After contact formation between VAP-A vesicles and PI4P tubes in the presence of OSBP or N-PH-FFAT, we observed significant remodeling of the vesicles, notably their flattening along the major axis of the tubes. When several tubes were involved in the contacts, vesicles flattened to face the tubes (Fig. 2b, Supplementary Fig. 3a, b).

We never observed VAP-A liposomes engulfing a tube; vesicles were either in reduced contact or formed tongues that partially

surrounded the tube (Fig. 3a, right, white arrow, supplementary Movie 4). In some cases, contact-forming vesicles bypassed high curvature areas of tubes (Fig. 3a, left). In the YZ the plane, the VAP-A membranes facing the tubes were flat and did not bend around the tube, whereas they could clearly curve outside the contact area (Fig. 3b). Moreover, VAP-A membranes in contact with the tubes did not show peaks of local deformation, in contrast to what has been reported for tricalbins by in situ cryo-EM of ER-plasma membrane contacts in yeast cells[14].

We hypothesized that VAP-A molecules concentrated in flat regions of MCS to avoid highly curved membranes. To test this, we analyzed how MCS formed when high curvature VAP-A vesicles were too tense to be remodeled (Fig. 3c). We reconstituted VAP-A at LPR 70 mol/mol and at 4 °C instead of 20 °C, leading to small and tense proteoliposomes that were homogeneous in diameter ($25 \pm 10$ nm; $n = 300$, 30 images). After incubation with N-PH-FFAT tubes, VAP-A vesicles showed no deformation upon binding along the tubes. Thus, VAP-A can form MCS even between highly curved membranes. However, the distance between the facing membranes was quite short, $15 \pm 2$ nm ($n = 52$, 4 images). Considering a lipid with a molecular surface of 65 Å$^2$, a vesicle with a diameter of 20 or 30 nm would contain 3800 or 8700 lipids, respectively, and therefore 52–114 VAP-A molecules at the given LPR. When the proteoliposomes were prepared at a high LPR (1400 lipids/VAP-A), leading to on average 2–4 VAP-A molecules per vesicles, the intermembrane distance was similar (ca. 15 nm). These observations suggest that, in an MCS between highly curved membranes, only a few homodimers of VAP-A were involved in the contact (as schematized in Fig. 3c). In contrast, when the VAP-A-containing membrane was deformable, proteins seemed to concentrate in a flat region of MCS, leading to larger separation between facing membranes, owing to the flexibility of VAP-A.

**3D models of membrane VAP-A/N-PH-FFAT contact site.** To determine the architecture of the contact site at the molecular scale, we performed sub-tomogram averaging. The principle consists of extracting sub-volumes containing identical proteins and associated membranes, followed by averaging them to increase the signal-to-noise ratio. We focused on MCS formed between VAP-A vesicles reconstituted at LPR 70 mol/mol and N-PH-FFAT-decorated PI4P tubes because they were more homogeneous than those formed in the presence of OSBP. Protein densities were visible in the contact areas between N-PH-FFAT-containing tubes and hemispherical or flattened VAP-A vesicles (Fig. 4a). In some cases, we observed continuous densities joining the two membranes. However, the distance between facing membranes of MCS varied, even in flat regions and we could not unambiguously attribute the densities to defined protein domains.

To circumvent these difficulties, we determined major 3D classes of MCS as a function of membrane separation (Fig. 4b). The different classes showed electron densities at three levels: (i) the tube membrane, which was well resolved, (ii) the vesicle membrane, which was more diffuse, and (iii) proteins between the two membranes. A plot of the electron densities perpendicular to the membrane plane is shown in Fig. 4c, in which we aligned the electron densities at the level of the lipid tube. The density of the VAP-A membrane was at increasing intermembrane distances from class 1 to 4. In contrast, a protein density was found at a constant distance of 5–6 nm from the tube membrane in all four classes. This density could be attributed to N-PH-FFAT as it was also present in the regions of the tubes not engaged in contacts. This observation suggests that the variable distance within the contacts resulted from the ability of VAP-A

molecules to adopt conformations of varying length and/or orientation.

We determined the architecture of the proteins present in class 3, which was the most homogeneous class by sub-tomogram averaging. During the analysis, we observed that the position and densities of N-PH-FFAT were more constant than those of VAP-A (Supplementary Fig. 4). Consequently, we treated the two proteins separately and obtained 3D models of VAP-A and NPH-FFAT at 19.6 Å and 9.8 Å, respectively (Supplementary Table 1).

VAP-A showed a rod-like shape of 14 nm long (Fig. 4d). The density attributed to the MSP domain could encompass a dimer confirming a dimeric organization of VAP-A in MCS. The N-PH-FFAT 3D model showed a T-shaped organization with a central stem extending 3 nm from the membrane and joining a 14 nm long and 2.7 nm wide rod-shaped structure parallel to the membrane (Fig. 4c). The axial symmetry was consistent with a dimer, in line with biochemical observations on OSBP and its N-PH-FFAT moiety[27,28].

Models of a homolog of the PH domain (aa 80–190 in OSBP) and of the complex between the disordered region G346-S379 containing the FFAT motif (aa 358–361) and VAP-A MSP are available[11,36]. In addition, we have shown that the N-terminal region of OSBP (aa1–80) is disordered[28]. However, there is no structural information on the region connecting the PH domain and the FFAT motif.

We performed structure predictions with several programs, which suggested similar secondary structures: two helices encompassing T204-S244 (H1) and E250-G324 (H2), followed by a long disordered region encompassing the FFAT motif (aa 325–408) (Fig. 5a, Supplementary Fig. 5b). We hypothesized that these two helices made the main contribution to the electron density of the T structure, whereas the PH domains would lie on the membrane. The PH domain and the unstructured parts should not visible at this resolution. To test this model, we expressed and purified an OSBP construct encompassing aa 199–324, hereafter termed Central Core (CC). The circular dichroism (CD) spectrum of this construct showed a 86% alpha-helical content, whereas gel-filtration chromatography suggested a dimeric organization (Fig. 5b, c).

In order to better understand the organization of the CC region of OSBP, we designed experiments to study the proximity of the N-termini and the C termini of this dimeric construct by intermolecular fluorescence resonance energy transfer (FRET) (Fig. 5d–h). We reasoned that if the aa chosen for fluorescent labeling on one chain was close to the cognate aa on the other chain, this should result in a high FRET signal.

We replaced the endogenous cysteines of OSBP CC (C224 and C276) with alanines and we introduced a cysteine either at the N-terminus (CC[V199C] construct) or at the C-terminus (CC[G324C]). All mutants behaved similarly by gel-filtration, suggesting no effect on the structure (Fig. 5c). After purification, each construct was labeled with a donor probe (AF-488), with an acceptor probe (AF-568), or with an equimolar ratio of the two dyes.

In the first set of experiments, we compared doubly labeled CC[V199C] and doubly labeled CC[G324C]. The first form showed a large FRET signal at 594 nm, suggesting proximity of the two N-termini in the dimer. The FRET peak vanished upon destabilization of the dimeric structure by SDS addition (Fig. 5d, e). Dually labeled CC[G324C] showed a 2.7-times smaller FRET peak, which was also abolished by SDS addition (Fig. 5e). The difference in FRET between the two mutants suggests that the C termini are more distant than the N-termini within the dimer.

In the second set of experiments, we mixed equal amounts of CC[V199C] labeled with AF-488 with CC[V199C] labeled with AF-568 (Fig. 5f). A slow FRET signal with a half-time of about

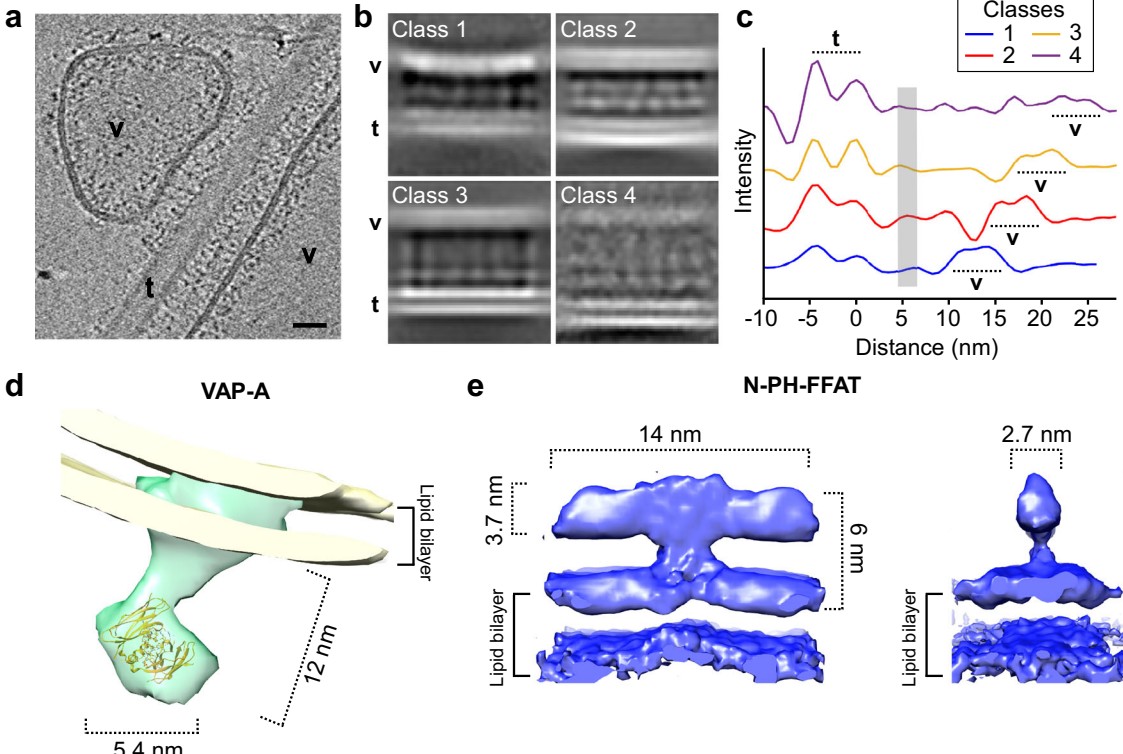

**Fig. 4 3D molecular architecture of VAP-A N-PH-FFAT in reconstituted membrane contact sites (MCS). a** Tomographic slice of VAP-A/NH-FFAT membrane contact site. Hemispherical vesicles (v) and large and flat vesicles (v) contact a tube (t) decorated with N-PH-FFAT bound to. The lipid bilayer of both vesicles is visible while the lipid bilayer of the tube is out of the plane. Tiny protein densities tether v and t. Bar = 25 nm. **b** Classes of membrane contact sites after 3D classification. Electron densities of the lipid bilayer and proteins are depicted in white. The distance between facing membranes increases from class 1 to 4. Class 1–3 correspond to MCS formed with VAP-A vesicles and class 4 with VAP-A ribbons. **c** 2D plot of electron densities perpendicular to MCS of classes 1–4. The outer leaflets of the tubes were aligned at 0 nm. The two lipid leaflets of the tube are well resolved while the VAP-A membrane is less defined and appears at increasing distance from class 1 to 4. In all classes, a protein density assigned to N-PH-FFAT domain is found at 5–6 nm from the tubes (gray line). **d** 3D volumes of VAP-A and **e** of N-PH-FFAT at 19.6 Å and 9.8 Å resolution, respectively. A representative tomographic slice or micrographs is shown from at least three replicates.

100 min appeared over time, indicating that monomer exchange between dimers occurred within several hours. Performing the same experiment with the CC[G324] mutant resulted in similar slow kinetics but with a threefold lower FRET signal confirming that the C termini are more distant than the N-termini within the dimer (Fig. 5g, h).

We generated 3D models of 195–335-OSBP with Robetta[37]. All models shared similar organization with H1 and H2 helices, H2 helix being continuous (e.g., model 1) or discontinuous (e.g., model 2) (Supplementary Fig. 5b). We used molecular dynamics (MD) simulations to analyze the dimer stability and found that model 1 formed a stable dimer (Fig. 5i). The distances between C-termini and N-termini along the MD trajectory (300 ns) are depicted in Fig. 5j. Within the dimer, the N-termini and the C-termini were close and distant, respectively, in agreement with the FRET experiments. This model was fitted in the EM envelop (Fig. 5k). Thus, this model seemed as the most likely but since the resolution was not enough to assign secondary structures, we cannot exclude that other models are possible. Finally, the stem of the T that connects to the membrane remains to be assigned.

## Discussion

When VAP-A is not engaged in contact, it appears as a flexible protein, extending its MSP domain at increasing distances that correlate with protein density. At a medium density, VAP-A shows various orientations with regards to the membrane plane, with its MSP domain exploring a 10 nm-thick region. At a very

high density that leads to ribbons-like membranes, the MSP extends to 17 nm away from the membrane (Fig. 1f). Structure predictions propose that two unstructured regions flank the predicted CC domain of VAP-A: region 135–171 between the MSP domain and the CC, and region 208–226 between the CC and the TM domain (Supplementary Fig. 5a). These regions, which are also present in VAP-B, might provide VAP-A/B dimers with bending and stretching capacity, allowing exploration of a larger space than a rigid molecule on the surface of the membrane (Supplementary Fig. 5c).

A most striking feature of VAP-A/B is their ability to interact with a myriad (currently estimated at ≈100) of cytosolic protein domains. VAP-A/B are in fact the only known receptors that anchor proteins on the ER surface. Importantly, the proteins that are recruited by VAP-A/B have very different domain organization and function. Some effectors are simply retained at the ER, others bridge the ER to various organelle membranes, as is the case for OSBP. For example, the Opi1 repressor interacts with the yeast VAP-A homolog as well as phosphatidic acid at the ER surface, whereas MIGA2 bridges ER to mitochondria[38,39]. These different molecular contexts might require different orientations and positioning of the MSP domain of VAP-A/B. Therefore, we propose that the structural flexibility of VAP-A/B enables a large spectrum of protein assemblies. This effect of protein flexibility is reminiscent of what has been observed in other membrane interfaces. Thus, flexible region of golgins, are essential for the capture of vesicles

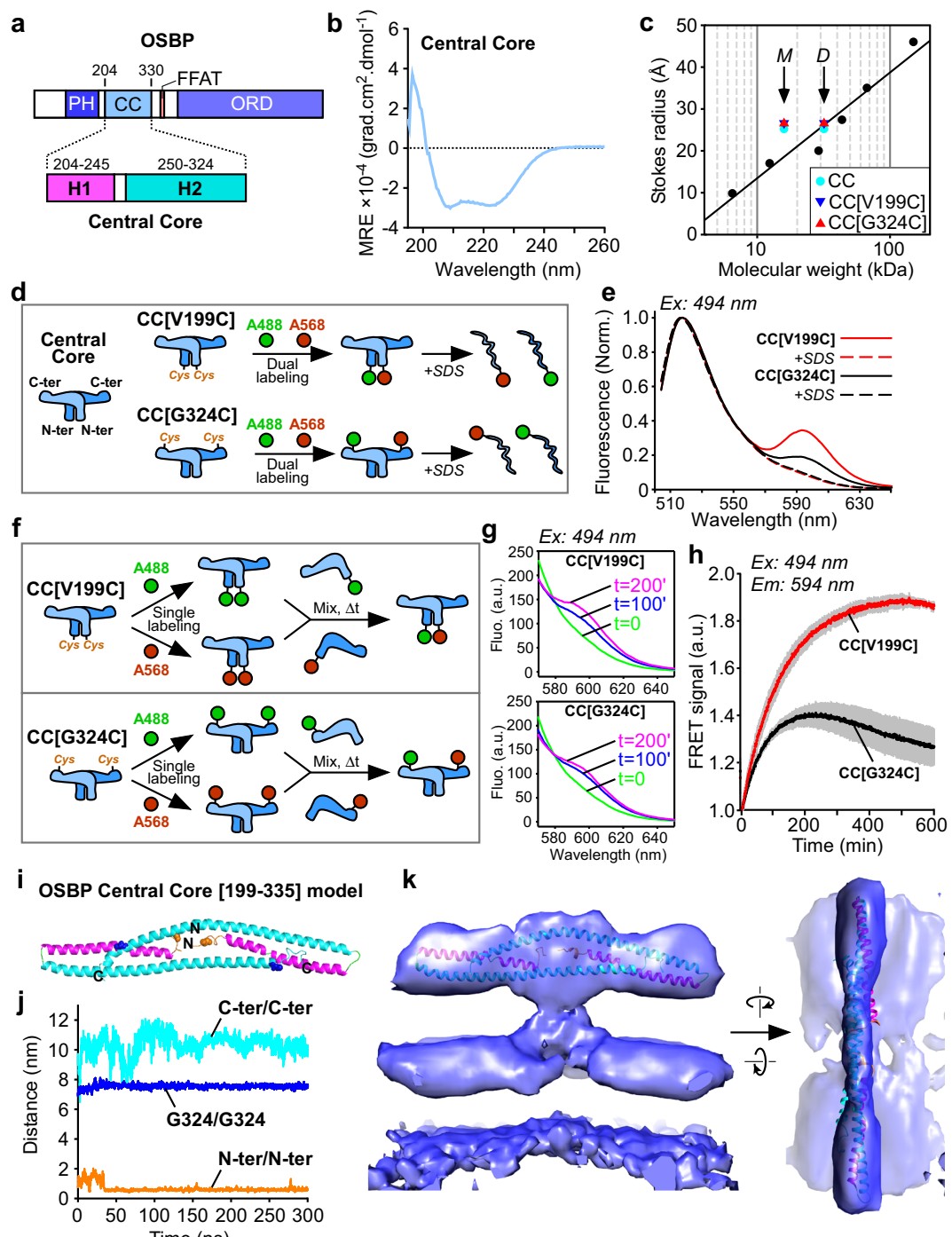

**Fig. 5 3D architecture of the central core domain of OSBP. a** Domain organization of OSBP with its central core (CC) region and the predicted H1 and H2 helices. **b** Circular dichroism spectroscopy of OSBP CC region. Circular dichroism CD spectra minima at 208 and 222 nm indicative of significant α-helical content. **c** Stokes radius versus MW of OSBP CC constructs (WT, light blue dot; V199C/C224A/C276A, blue triangle; C224A/C276A/G324C, red triangle) and standards (black dots) as determined by gel filtration. The arrows indicate the expected positions for monomers (M) or dimers (D) according to their MW. **d** Principle and (**e**) spectra of dual labeled AF-488 and AF-568 OSBP CC on the N-ter (V199C) or the C-ter (G324C) region analyzed by fluorescence resonance energy transfer (FRET). FRET signal vanished after dissociation of dimers of OSBP CC region with SDS. **f** Principle of OSBP CC monomer exchange experiment as analyzed by FRET **g** spectra and **h** kinetics analyses of monomer exchange between mono labeled AF-488 and mono labeled AF-568 OSBP CC region. Labeling was performed and a Cys introduced either on the N-ter (V199C) or the Cter (G324C) of a construct having endogenous Cys substituted to Ala (C224A/C276A). **i** Structure of the dimer OSBP CC after 300 ns MD simulation colored by chains. **j** Plot of the distances between C-termini, N-termini, and G324/G324 along the MD trajectory (300 ns). **k** Fit of the dimer OSBP CC after 300 ns MD simulation in the 3D EM model of N-PH-FFAT. A representative result is from at least three replicates.

on the surface of the Golgi[40]. In Ist2, a yeast MCS tether, a long intrinsically disordered region is responsible for membrane tethering, spatial freedom and protein recruitment[41,42]. Model studies of the interaction between streptavidin-coated vesicles and flat surfaces coated with either flexible PEG-biotin or rigid casein–biotin ligands have shown that molecular flexibility can increase the rate of adhesion between two surfaces by three orders of magnitude[43].

When VAP-A is engaged in a MCS with N-PH-FFAT, 3D classification reveals that, again, VAP-A is variable in length and determines the separation between membranes; in contrast, N-PH-FFAT displays a fixed length (Fig. 4c). The intermembrane distance increases with the concentration of VAP-A in MCS from 10 nm when a few VAP-A molecules are reconstituted in small vesicles to 30 nm when VAP-A molecules are densely compacted in ribbons (Figs. 2c, 3d).

VAP-A forms MCS with OSBP or its membrane tethering moiety N-PH-FFAT. However, less VAP-A molecules are engaged within MCS formed with OSBP than N-PH-FFAT as shown by the shorter intermembrane distance suggesting that the bulky ORDs decrease the accessibility of MCS to VAP-A (Fig. 2D). VAP-A is thus strikingly different from a rigid tethering protein that imposes a fixed distance between the facing membranes. In that case, as shown by model mixtures of binding and non-binding proteins of variable lengths, a difference in length of only 5 nm of the non-binding proteins is sufficient to drive exclusion from MCS[44]. The much larger range of intermembrane distance in VAP-A mediated MCS suggests that VAP-A adapts its density and dimension to the size of its interactor within a contact area (Supplementary Fig. 5d).

At ER-Golgi MCS, VAP-A recognizes several LTPs including OSBP, CERT, Nir2, and FAPP2, which deliver cholesterol, ceramide, PI, and glucosylceramide to TGN, respectively[22]. They all encompassed a PH or a LNS2 (Nir2) domain that binds PI4P at the Golgi and a FFAT motif. As a result, these LTPs could bind simultaneously to both organelle membranes and thus act as bridging factors. It is not known if VAP-A has different affinity for these different LTPs and if all LTP could coexist within the same MCS. However, our results suggest that the flexibility of VAP-A could support the coexistence of different partners involved in lipid transfer within the same MCS. Similarly, it has been shown by confocal microcopy that the phosphoinositide transfer protein Nir2 is recruited at ER-PM neuronal contact sites made by VAP-A and the potassium channel Kv2:1. Nir2 and Kv2:1 bear a FFAT and FFAT-like motif, respectively. Such tripartite ER-PM junction is proposed to play a role in the phophoinositide homeostasis in brain neurons[45].

During in vitro formation of MCS, large VAP-A vesicles spread along the major axis of the PI4P containing tubes until they were no longer deformable (Fig. 3a). This suggests the participation of several VAP-A/N-PH-FFAT tethering complexes in an MCS, which could compensate for the moderate affinity (Kd 2 μM) of FFAT-bearing proteins for MSP[11]. On the other hand, we observed no wrapping of VAP-A vesicles around the highly curved tubes, which would require strong interaction between VAP-A and N-PH-FFAT or OSBP. When membrane curvature is too high, like in small and tensed proteoliposomes, the separation is small and independent of the initial density of VAP-A in the proteoliposomes (Fig. 3c). Thus, the high curvature in small proteoliposomes prevents VAP-A molecules from forming a dense protein area as observed in flat membranes (Supplementary Fig. 5d). In situ cryo-EM has shown that Scs2, the yeast ortholog of VAP-A, also accumulates in ER sheets, whereas tricalbins accumulate in tubular ER[15,46]. This difference was attributed to the cylindrical shape of Scs2 with a single TM and an elongated

cytosolic domain as compared with the conical shape of tricalbins, which insert in the membrane via a hairpin and display bulky C2 domains.

Once in flat regions, how VAP-A concentrate in MCS remains an open question. Clusters of VAP-A have been reported as a consequence of domains of Kv2.1 potassium channel in the PM and binding of VAPs to form ER-PM contact site[35]. Concentration of VAP-A might also result from geometrical constraints: when an elementary VAP-A/N-PH-FFAT pair forms, similar pairs should concentrate in the vicinity as the intermembrane distance is optimal for assembly. VAP-A might also concentrate in lipid nano-domains present in ER as reported recently for other MCS[47]. However, our liposome composition was not prone to form membrane domains. Whatever the process of concentration of VAP-A, our results suggest that it should have an impact on the separation of facing membranes within MCS.

The tethering region of OSBP is organized as a T-shaped with a 14 nm elongated domain parallel to the membrane and a short ≈3 nm stem. The resolution is not enough to assign a specific domain of OSBP but since it connects to the membrane, it probably contains part of the two PH domains. Biochemical analysis indicates that the region between the PH domain and FFAT motif is dimeric, alpha-helical and has its N-termini very close together, whereas its C-termini are more distant (Fig. 5). Regardless the exact atomic structure, which will require further studies, the consequence of this organization is that the tethering and lipid transfer moieties of each monomer are separated by a large distance, akin to the jib of a tower crane. Moreover, the two ORDs of the OSBP dimer should be well separated from each other, which might facilitate their independent movements.

Following the T domain, a long sequence G324-R408 is predicted to be unstructured (Supplementary Fig. 5b). The first part of this sequence, G324-N357, i.e., 31 aa before the FFAT motif, might extend up to 10 nm beyond the T domain to interact with VAP-A (considering a 3.5 Å step per residue). As a result, the orientation and position of the T domain relative to VAP-A might vary. This is what we observed when analyzing a single 3D class as defined by the fixed intermembrane distance: the VAP-N-PH-FFAT complexes were not at fixed positions thus limiting the resolution of the whole complex. The second part of the sequence A362-R408 (47 aa), which unifies the FFAT motif to the ORD, could further extend 14 nm away and should define the action range of the ORD between the MSP of VAP-A and the two facing membranes. This ball-in-chain geometry could allow a movement of up to 28 nm of the ORD between the ER and TGN membranes, which might be enough to exchange lipids between the two facing membranes, considering that MCS at the ER TGN interface have an intermembrane distance of 5–20 nm[48]. In addition, the flexibility of VAP-A as revealed here by cryo-EM could further facilitate the movement of the ORD.

We have previously shown that the intrinsically disordered N-terminal region of OSBP controls its density under confined conditions, which facilitates lateral diffusion of proteins within MCS[28]. Here, we highlighted the flexible properties of VAP-A and the ordered and disordered domains of OSBP. Thus, the molecular actors of VAP-A/OSBP MCS carry interfacial recognition information, and also contain regions that are intrinsically disordered and flexible, which is likely crucial for MCS dynamics and supramolecular assembly.

## Methods
**Microbe strains**. Recombinant VAP-A proteins were expressed in C41 (DE3) *E. coli* strain, which were grown in 2×YT medium at 30 °C. Recombinant OSBP proteins were expressed in Bl21 (DE3) *E. coli* strain, which were grown in LB/ Ampicilin medium at 37 °C and induced at 16 °C.

**Reagents**. Egg PC, brain PS, brain PI(4)P, C24:1 Galactosyl(ß) Ceramide, Texas Red 1,2-dihexadecanoyl-sn-glycero-3-phosphoethanolamine, triethylammonium salt (Tx-DHPE) were from Avanti Polar Lipids. Anapoe-X-100 (Triton X100), DDM was purchased from Anatrace. Peptidol:HS-PEG capped gold nanoparticles of 6 nm were prepared by Dr. L. Duchenes[49]; 50 mesh BioBeads were purchased from Bio-Rad and prepared according to ref. [50]. Lacey carbon electron microscopy grids were purchased from Ted Pella (USA).

**Plasmids and proteins expression**. Human VAP-A (8-249) fragment was cloned into pET-His6-TEV-LIC then sub-cloned into pET16b for inducible expression as N-terminal His-StrepII-TEV-Vap-A. Vap-A expression was performed in C41 (DE3) E. coli strain. Full-length VAP-A expression was performed in presence of 0.5 mM IPTG for 4 h at 30 °C.

OSBP central region (198–324) were PCR amplified as either (V199C/C224A/C276A) or (C224A/C276A/G324C) mutants and sub-cloned via BamH1/Xma1 sites into pGEX-4T2 for expression as GST-tag proteins. Proteins were expressed in Bl21gold E. coli after induction with 1 mM IPTG and overnight culture at 16 °C. The list of all primers is supplied in Supplementary Table 2.

**Proteins purification**. VAP-A: VAP-A-expressing cells were collected and passed twice through a Cell Disrupter at 2k bar pressure. The resulting cell lysate was centrifuged to purify membranes. Membranes were solubilized for 2 h at 4 °C with DDM:protein ratio of 2.5 (w/w) and then centrifuged to collect solubilized proteins. VAP-A was purified on Strep Tactin beads (IBA superflow sepharose) in batch. Eluted proteins were incubated overnight with TEV protease to eliminate the double tag, followed by purification on a Superdex 200 (GE healthcare) size exclusion chromatography in Tris 50 mM pH 7.5, NaCl 150 mM, DDM 0.03 mol/l, 10% glycerol. Purified VAP-A was collected at concentration ranging from 1.5 to 2 mg/ml without any additional concentration step, flash frozen in liquid nitrogen and stored at −80 °C.

OSBP and N-PH-FFAT were purified according to refs. [23,28]. GST-OSBP (199–324) mutants expressing cells were pelleted and resuspended in 50 mM Tris (pH 7.5), 300 mM NaCl, 5 mM DTT buffer. The bacteria were lysed through a Cell Disrupter at 2k bar pressure and cell lysate was ultracentrifuged. Soluble GST-tagged proteins were first purified on Glutathion-Sepharose 4B (GE healthcare) beads, eluted after thrombin clivage, and then further purified on a MonoS HR5/5 column (GE healthcare). Proteins were eluted from the column with a 0 to 1 M NaCl gradient (25 column volume) in 25 mM MES (pH 6.0), 3 mM DTT buffer. Protein fractions were pooled, concentrated on Amicon Ultra 4 (10 kDa cutoff), injected on a Superpose 12 HR 10/30 column (GE healthcare) and eluted with Tris 25 mM pH 7.5, NaCl 120 mM, supplemented with an antioxydant (3 mM DTT or 0.5 mM TCEP). Proteins fractions were pooled, concentrated on Amicon Ultra 4 (10 kDa cutoff), and supplemented with 10% glycerol before flash freezing in liquid nitrogen and stored at −80 °C.

**Analytical gel filtration**. Purified proteins (100 µl, 10 µM) were applied to a Superdex 75™ column (GE Healthcare) and eluted at a flow rate of 0.5 ml min⁻¹ in 25 mM Tris pH 7.5, 120 mM NaCl, and 1 mM DTT. The column was calibrated using the following standards (MW/Stokes radius): Bovine serum albumin (67 kDa/3.5 nm), Ovalbumin (43.5 kDa/2.7 nm), Carbonic anhydrase (25 kDa/2.1 nm), Cytochrome C (12.4 kDa/1.7 nm) and Aprotinin (6.5 kDa/0.9 nm). The elution volume and Stokes radius of the standards were used to establish a first calibration curve, from which the Stokes radius of the OSBP constructs were determined. Thereafter, we plotted the Stokes radius as a function of MW for both protein standards and for OSBP constructs.

**Fluorescent labeling**. DTT present in OSBP (199–324) CYS mutants was removed by buffer exchange on a NAP-5 column (GE healthcare). Proteins were incubated with a 10-fold mol excess of Alexa Fluor (AF) dyes (AF-488 or AF-568) for 1 min at room temperature and then 30 min on ice. For dual labeling, proteins were similarly treated but with a fivefold mol excess of each Alexa dye (AF-488 and AF-568). In this particular case, TCEP-containing protein were diluted to reduce TCEP concentration below Alexa dye concentration (1.5 µM total dye/1 µM TCEP). Reaction was stopped by addition of a 100-fold excess of L-cystein. Excess unbound dye and L-cyst were removed by elution on a NAP-5 column (GE healthcare). The extent of AF-labeling was estimated by spectrophotometry and by SDS-PAGE analysis on a gel imaging system adapted for fluorescent proteins.

**CD**. Circular dichroism spectroscopy was performed on a Jasco J-815 spectrometer at room temperature with a quartz cell of 0.05 cm path length. Each spectrum is the average of 10 scans recorded from 195 to 260 nm with a bandwidth of 1 nm, a step size of 0.5 nm and a scan speed of 50 nm min⁻¹. The purified construct OSBP (198–324) was dialyzed in Slide-A-Lyzer dialysis cassette (Thermo) against KCl 150 mM, Tris 10 mM pH 7.5, DTT 1 mM, and was used at 48 µM. Control spectra of buffer alone was subtracted from the protein spectra.

**Reconstitution of VAP-A in proteoliposomes**. VAP-A was reconstituted in lipid mixture made of 9.5/0.5 w-w egg PC/brain PS by detergent removal using BioBeads

(see refs. [51,52]). Briefly, DDM solubilized VAP-A was mixed at room temperature with egg PC/PS liposomes solubilized in Triton X100, detergent/lipid ratio of 2.5 w: w, in 50 µl volume. Detergent was removed by additions of wet BioBeads at Bio-Beads/detergent ratio of 20 w:w 2 h/20 w-w 1 h/20 w-w 1 h at room temperature or at a ratio of 40 w-w for 2 h and a second addition of 20 w-w for two more hours at 4 °C. This amount of BioBeads allowed a complete detergent removal[50]. Reconstitution buffer was 50 mM Hepes pH 7.4 and 120 mM potassium acetate (HK buffer), 1 mg/ml lipid, 2.5 mg/ml Triton X100. Protein was added at LPR from 70 to 2800 mol/mol. After reconstitution, proteoliposomes were kept at 4 °C for a maximum of 2 days before use.

**VAP-A incorporation into proteoliposome**. VAP-A was reconstituted as described above in 9.5/0.5 w:w egg PC/brain PS and 0.5 w/w of fluorescent Tx-DHPE, at LPR1400 mol/mol. Then 100 µl of VAP-A proteoliposomes were mixed with freshly prepared ice cold 50% w/w sucrose in gradient buffer (Tris pH 7.4, 50 mM, NaCl 150 mM) to get final 575 µl of 30% sucrose concentration fraction and loaded at the bottom of a gradient tube (2.5 ml, Ref. 347357 Beckman Coulter). Then 575 µl fraction of 20, 10, and 5% sucrose concentration were carefully loaded one after the other to create a discontinuous gradient. Gradient preparation was performed at 4 °C to allow a good separation of sucrose fractions and avoid any mixing.

**PI4P transport**. Golgi-like liposomes were prepared with 64 mol % egg PC, 17% liver PE, 8% liver PI, 4% PI4P, 5% brain PS supplemented with 2 mol % Rho-PE and 4 mol % PI4P. VAP-A proteoliposomes were reconstituted at LPR 1450 mol: mol in 95% egg PC/5% brain PS, supplemented with 10% cholesterol. Golgi-like liposomes (250 µM lipids) were incubated at 37 °C in HKM buffer with 0.3 µM NBD-FAPP1 PH domain (NBD-PH) and with VAP-A proteoliposomes (250 µM lipid–0.17 µM VAP-A) in a cylindrical quartz cuvette (total volume 600 µl). The sample was continuously stirred with a magnetic bar. PI(4)P transport was followed by measuring the NBD emission signal at 530 nm (bandwidth 10 nm) upon excitation at 460 nm (bandwidth 1 nm) in a JASCO fluorimeter. At the indicated time, 0.1 or 0.2 µM OSBP was added.

**FRET measurement**. Fluorescence resonance energy transfer (FRET) was measured in a JASCO fluorimeter. Dual AF-labeled constructs were used at 200 nM. Spectra emission was recorded from 500 to 650 nm (5 nm bandwidth) upon excitation at 494 nm (1 nm bandwith) in the absence or in the presence of 0.5% SDS. Three spectra per condition were normalized at maximal direct fluorescence of the donor dye. For measurement of monomer exchange within dimers, we mixed equimolar amounts (100 nM) of AF-488 and AF-568 CC constructs. Spectra were measured every 10 min for CC[V199C] and every 5 min for CC[G324C] (total 20 spectra each). Alternatively, emission was continuously recorded at 594 nm (5 nm bandwidth).

**Preparation of galactocerebroside tubes**. Galactocerebroside (Galcer) tubes doped with PI4P were prepared according to[34]. In brief, a mixture of Galcer, egg PPC, brain PS, brain Pi4P was dried under vacuum. Galcer/egg PC/PS/Pi4P (80/10/5/5) tubes were formed at room temperature after resuspension of the dried film at 5 mg/ml lipid concentration in HK buffer followed by five cycles of 10 min vortex, 2 min at 40 °C. GalCer tubes were aliquoted and stored at −20 °C.

**Formation of in vitro MCS**. VAP-A/N-PH-FFAT and VAP-A/OSBP MCS were formed directly on the cryo-EM grid at room temperature. VAP-A proteoliposomes were diluted at 0.05 mg/ml in HK buffer. Galcer tubes were diluted at 0.15 mg/ml and mixed with NPH-FFAT or OSBP at 70 lipids protein mol/mol ratio.

A 2 µl drop of NPH-FFAT or OSBP tubes was immediately loaded on the grid and let incubate for ~1 min, followed by the addition of 2 µl VAP-A liposomes. After 30 seconds incubation, grid was blotted and immediately frozen. For cryo-ET experiments, 6 nm gold beads were added before plunging. This protocol was used for all reconstitutions of VAP-A with LPR ranging from 70 to 2800 mol/mol.

**Cryo-EM experiments**. Lacey carbon 300 mesh grids (Ted Pella, USA) were used in all cryo-EM experiments. We found that lacey networks with holes ranging from 20 nm to few microns entrapped different reconstituted material more than calibrated Quantifoil grids. In all experiments, blotting was carried out on the opposite side from the liquid drop and plunge frozen in liquid ethane (EMGP, Leica, Germany). Samples were imaged using different electron microscopes. Data and 2D images depicted in Fig. 1c, Supplementary Fig. 1b–f, Supplementary Fig. 2a–e, Fig. 3c and Supplementary Fig. 3d were acquired with a Tecnai G2 (Thermofisher, USA) Lab6 microscope operated at 200 kV and equipped with a 4k × 4k CMOS camera (F416, TVIPS). Image acquisition was performed under low dose conditions of 10 e⁻/Å² at a magnification of 50,000 or 29,500 with a pixel size of 2.1 or 3.6 Å, respectively. Plots depicted in Figs. 1e, 3d were derived from images taken with the Tecnai G2 electron microscope. Data and 2D images depicted in Figs. 1e, f, 2b, plot Fig. 2b were acquired with a Polara (Thermofisher, USA) FEG 300 kV microscope with a K2 Gatan camera in counting mode with 40 frames during 6 sec, total dose 80 e⁻ and at 0.96 Å/px.

**Cryo-ET data acquisition**. Tilt series from samples of VAP-A/OSBP and VAP-A/N-PH-FFAT were acquired with a Titan Krios microscope (Thermofisher, USA) operated at 300 keV and equipped with a Quantum post-column energy filter and a Gatan K2 Summit direct detector. Tilt series, consisting on 41 images, were collected under the dose-symmetric scheme[53] in the angular range of ±60°, with angular increment of 3°, defocus range between −2.0 and −4.8 μm and pixel size of 1.7 Å. Every image was composed of 10 frames with a total dose of $3.5 \, e^-/\text{Å}^2$. The total dose for each tilt series was $143.5 \, e^-/\text{Å}^2$. High-resolution movies were aligned with MotionCor2[54], and the average defocus was estimated with CTFPlotter and corrected with CTFphaseflip[55]. The resulting micrographs were dose-filtered according to ref. [56] by means of Matlab scripts (https://github.com/C-CINA/TomographyTools).

During optimization of cryo-ET samples, we observed that protein A gold beads bound with high affinity to VAP-A. To reduce binding, PEG-coated gold beads were added to samples employed for cryo-ET[49]. We observed that much more gold beads were at the proximity of MCS formed by OSBP than with N-PH-FFAT. Cryo-ET (see Fig. 3e) revealed that less molecules of VAP-A were engaged in contacts with OSBP and thus more accessible for binding gold particles. However, this prevents further analysis by sub-tomogram averaging of OSBP-VAP-A MCS. Data collection are presented Supplementary Table 1.

**2D image analysis**. In 2D cryo-EM images, the lipid bilayer of the vesicles or the tubes are identifiable and serve as landmarks for distance measurements. In the plot Fig. 1e, the lengths of the extramembraneous domains of VAP-A are reported from the electron densities extending at the largest distances from the membrane. A number of 50 vesicles ($n = 78$ measurements), 45 ribbons ($n = 421$ measurements) and 6 onions ($n = 78$ measurements) have been analyzed. In the case of onions we consider VAP-A length as half of the intermembrane distance although we have no details on the molecular organization of VAP-A homotethers. In Fig. 2b, contact areas are identified by the presence of deformed and spread vesicles in contact with tubes and electronic densities between membranes. In the case of spherical vesicles with low density of VAP-A, vesicle involved in contacts are identified by their proximity, <30 nm, to tubes. In addition, to avoid selecting vesicles close to tubes owing to too high a concentration of material in the holes, only areas where tubes and vesicles are sparsely distributed in the holes of the lacey are considered. Distances between facing membranes were measured every 20 nm along the contact zone. A total of 90, 122, and 208 distances of contact made between tubes and spherical, hemispherical, and angular-shaped vesicles were analyzed and depicted in the plot Fig. 2c. In the plot shown in Fig. 3c, a number of 52, 240, 491, 152 small vesicles have been analyzed for LPR 70, 175, 350, and 1400 lipid/VAP-A mol/mol, respectively.

**Sub-tomogram averaging**. Tubes containing PI(4) N-PH-FFAT and vesicles reconstituted with VAP-A were identified and segmented in Dynano[57,58]. We performed a multireference analysis in Dynamo, using as initial seeds averages of four random combinations of particles; classes representing different inter-membrane lengths arise spontaneously. The iterative process stopped after no swapping of particles between classes was observed. The distance between tubes and vesicles was measured and particles were classified in classes corresponding to distance ranges of 0–5 nm, 5–10 nm, 10–15 nm, 15–20 nm, 20–25 nm, and 25–30 nm. Further analysis was performed with the most homogeneous class 20–25 nm. The full analysis is described in supplementary information (Supplementary Fig. 4). In brief, sub-volumes extracted from the center of the contact zone were aligned using a cylindrical mask covering both membranes and the contact region. A new round of alignment, in which the initial reference was low-pass-filtered to 48 Å, was run using a mask that covered the contact zone, where density corresponding to protein complexes was already observed. Features of the complex VAP-A with N-PH-FFAT were resolved but attempts to align them to higher resolution were unsuccessful, likely owing to flexibility of the complexes. Thus, individual components N-PH-FFAT and VAP-A of the complex were analyzed separately. Angular search and shifts were gradually decreased while resolution was gradually increased by moving from bin4 to bin2 sub-volumes and by moving the low-pass filter towards higher resolution. The overall resolutions for VAP-A and of NPH-FFAT as determined by splitting and analyze the data set at step 5 of the work-flow from the gold standard Fourier shell correlation (FSC) processing using FSC0.134 was 19.6 Å and 9.8 Å, respectively (Supplementary Fig. 4). For NPH-FFAT, a further inspection of the local resolution computed by the program blocres (BSoft)[59] pointed at a distribution of resolutions in the range of 9–18 Å (Supplementary Fig. 4). Data processing are presented Supplementary Table 1.

**Structure prediction and modeling of proteins**. Secondary structure predictions of human VAP-A (Q9P0L0) and N-PH-FFAT, i.e., 1-408-OSBP have been performed using Phyres2, PSIPRED 4.0[60,61].

3D models of 195–335 OSBP were generated using the homology method provided by Robetta, the protein structure prediction server[62] (Supplementary Fig. 5b).

**MD simulation of dimer model**. The 3D model of 195–335 OSBP (Model 1, Supplementary Fig. 5b) generated from Robetta was duplicated to create two chains and positioned with on average 9 Å distance between them with Pymol software[63]. MD simulations were performed with GROMACS 2019.4[64] with the CHARMM36 force field[65]. The dimer was centered in the cubic box, at a minimum distance of 2.5 nm from box edges. Solvent molecules were added in the coordinate file. The TIP3P water model configuration was used (Jorgensen et al., 1983). Na and Cl ions were added to neutralize the simulation box and at a minimum concentration of 120 mM. The total number of atoms (protein + solvent + counterions) was 958,802.

Energy minimization was performed using the steepest descent minimization algorithm for the subsequent 50,000 steps. A step size of 0.01 nm was used during energy minimization. A cutoff distance of 1 nm was used for generating the neighbor list and this list was updated at every step. Long-range electrostatic interactions were calculated using the particle mesh Ewald summation methods[66]. Periodic boundary conditions were used. A short 100 ps NVT equilibration was performed. During equilibration, the protein molecule was restrained. All bonds were constrained by the LINear Constraints Solver (LINCS) constraint-algorithm[67]. A short 100 ps NPT equilibration was performed similar to the NVT equilibration. During the production run, the V-rescale thermostat and Parrinello–Rahman barostat[68] stabilized the temperature at 300 K and pressure at 1 bar, respectively. The simulations were performed for 300 ns and coordinates were saved every 100 ps. The MD analysis (distance between Nter/Cter) were performed using Gromacs utilities. Similar procedure was performed with the 3D model of 195–335 OSBP (Model 2, supplementary Fig. 5b) without leading to a stable dimer.

**Visualization**. Rendering of 3D density maps is performed with UCSF Chimera[69]. The threshold of intensity was defined as it corresponded to a point where the intensity distribution in the map showed a change of derivative. Rigid-body docking of the crystallographic structure of the dimer of VAP-A MSP (PDB ID code 1Z90) and of the computed (195–335)-OSBP dimer into the cryo-EM density maps was done with Chimera. The dimer of MSP was selected over the monomer based on the correlation calculated by Chimera, 0.93 for the dimer vs 0.89 for the monomer, after fitting the model into the map by simulating a map/calculating a simulated map at 20 Å. The lower correlation was related to the lower mass of the monomer compared with that of the calculated map. A tetramer was also fitted into the map but its volume surpassed that of the calculated map. The estimated correlation was 0.80 in this case.

**Reporting summary**. Further information on research design is available in the Nature Research Reporting Summary linked to this article.

## Data availability

Data supporting the findings of this paper are available from the corresponding authors upon reasonable request. A reporting summary for this Article is available as a Supplementary Information file. Source data are provided with this paper.

We have deposited the EM maps and models to the Electron Microscopy Data Bank with accession codes:

https://www.ebi.ac.uk/pdbe/entry/emdb/EMD-11455
https://www.ebi.ac.uk/pdbe/entry/emdb/EMD-11438
https://www.ebi.ac.uk/pdbe/entry/emdb/EMD-11402
https://www.ebi.ac.uk/pdbe/entry/emdb/EMD-11427
https://www.ebi.ac.uk/pdbe/entry/emdb/EMD-11399
https://www.ebi.ac.uk/pdbe/entry/emdb/EMD-11376

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

## Acknowledgements

We are grateful to A. Bertin for 2D image analysis, P. Cuniasse and P. Sens for fruitful discussions, D. Woolfson for feedback on CC analysis, N. Charmel for the picture of Figures 2A and 3C, L. Duchesne for providing PEG-coated gold beads and A. Copic for critical reading. We thank G. Schoehn and the EM facilities at the Grenoble Instruct-ERIC Center (ISBG; UMS 3518 CNRS CEA-UGA-EMBL) with support from the French Infrastructure for Integrated Structural Biology (FRISBI; ANR-10-INSB-05-02). We are deeply grateful to W. Hagen for the quality of the cryo-electron tomography data acquired at the cryo-electron microscopy platform of the European Molecular Biology Laboratory (EMBL) in Heidelberg. We thank the Cell and Tissue Imaging core facility (PICT IBiSA), Institut Curie, member of the French National Research Infrastructure France-BioImaging (ANR-10-INBS-04). This work was supported by CNRS, Institut Curie, and the Agence Nationale de la Recherche (ANR-15-CE11-0027-02). This work was also supported by iNEXT, project number iNEXT PID7315, funded by the Horizon 2020 program of the European Union. E. de la Mora was funded by a grant from ANR (ANR-15-CE11-0027-02) and by a grant from the Labex CellTisPhysBio (ANR-11-LABX-0038, ANR-10-IDEX-0001-02). The Antonny lab is supported by a grant from the Fondation pour la Recherche Médicale (Convention DEQ20180339156 Equipes FRM 2018), by the LABEX signalife (ANR-11-LABX-0028-01) and by the university Côte d'Azur (IDEX académie 4—masters environnés). D.C.D. acknowledges support from Human Frontier Science Program (HFSP) grant RGP0017/2020 and the Swiss National Science Foundation (SNF) grant 205321_179041.

## Author contributions

Conceptualization: M.D., B.M., B.A., and D.L.; purification of LTPs and lipid transfer assays: J.B., J.P., B.M. and of VAP-A, M.D., and J.M. Cryo-EM, cryo-ET: A.D.C., M.D., and D.L. Biochemical analysis: J.B., B.M., B.A., and M.D. Image analysis: E de la M, D.C.D. Modelisation: D.L., R.G. Writing—original draft: D.L. and M.D.; writing—review & editing: D.L. and B.A. Funding acquisition: B.M., B.A., and D.L.

## Competing interests

The authors declare no competing interests.
