## [Peer Review File · Nature Communications]

Reviewer #1 (Remarks to the Author):

In this manuscript, entitled 'Nanoscale architecture of a VAP-A-OSBP tethering complex at membrane contact sites', de la Mora and colleagues reconstitute membrane contact sites (MCS) in vitro and use cryo-electron tomography to visualise the MCS structure.

The reconstitution is performed by adding a pair of tethering proteins, VAP-A and the OSBP FFAT domain to vesicular and tubular membranes respectively.

The authors observe interaction between these two membranes, and this causes deformation of the VAP-A containing vesicles.

They observe different types of interactions/deformations, and find that the inter-membrane distance varies between them, suggesting different conformations of the VAP-A protein.

Subtomogram averaging is used to derive low-resolution structures of the two protein components and to propose a model of their architecture at MCS.

The in vitro reconstitution of MCS is a nice achievement, as well as the observation of heterogeneity that can be explained by VAP-A flexibility. However, I find many of the conclusions beyond this are not well supported by the data.

Specific criticisms are listed below:

Major comments:

1. Molecular weight and oligomerisation states are derived from gel filtration profiles. SEC is not a reliable methodology to determine molecular weight, and such conclusions should be supported by other methods e.g. SEC/MALS.
2. If there is an interaction between OSBP and VAP-A this should be seen by floatation of OSBP when added to the experiments discussed in Figure 1C. Has this test been done? It would further validate that the MCS formation seen by cryo-EM is due to specific protein-protein interaction.
3. On page 7, the authors discuss 'dark dots' and assign them to specific domains. There is no reason to think that certain domains would be darker than others and this discussion should be removed.
4. On page 7, the authors discuss that the elongated shape is consistent with high molecular weight size-exclusion chromatography. See point 1, this should be removed.
5. In Figure 2, the authors suggest that the full-length OSBP imposes shorter space between opposing membranes because it is less packed and therefore VAP-A adopts an elongated conformation. Given that difference distances (attributed to different VAP-A conformations) are also seen in round vesicles versus flattened ones versus ribbons, this means there should be a correlation. I.e. MCS reconstituted with full-length OSBP should be enriched in vesicles over ribbons, and viceversa? Do the authors observe this and if not, why?
6. How is the classification shown in Figure 4B performed? This is missing from the method and it is not clear whether it is a bona fide classification, or whether the data was simply 'divided' in groups depending on distance measured? Why 4 classes? These details are significant and must be included.
7. How was the threshold established for the VAP-A structure in figure 4D? The authors choose an (arbitrary as far as I can see) threshold, fit a dimer, and derive information about the oligomeric state of the protein. They could have equally fit a dimer and assigned a threshold to match its density. Given the absence of recognisable features I question the reliability of size measurements of VAP-A.
8. Similarly, for the N-PH-FFAT density in Figure 4E: how was the threshold decided?
9. For both structures, I have concerns over their validation. I am sorry to say that the density of N-PH-FFAT does not appear to reach a resolution anywhere close the claimed 9.8Å. While I commend the inclusion of supplementary Figure 4, I am concerned about noise alignment, seeing how the

density abruptly disappears outside the masks used. It is not clear in the methods at which point the dataset was split in two halves: before alignment or only prior to FSC calculation? Is the elongated shape of N-PH-FFAT also seen in the raw data? How do the subunits orient with respect to each other and to the membrane in 3D? Have the authors checked that these long rods do not clash with each other, or the VAP-A with the N-PH-FFAT densities?

10. Finally, I also have concerns regarding the reliability of the model fitted in the N-PH-FFAT density. Firstly, the authors assume the ordered part of the structure is composed by two helices based on secondary structure predictions. Secondly, they perform FRET experiments to establish N-termini are in close proximity (assuming dimerization, which is only suggested by SEC analysis). Models are next built with Robetta, and chosen based on stability as assessed by Molecular Dynamics. All in all, this process makes a number of assumptions that are not experimentally validated.

Minor comments:

1. References are missing throughout the first part of the introduction
2. Page 4, 1st sentence: "this contact site". Which?
3. Same sentence as above. While it becomes clear later that OSBP contains FFAT motif, this should be spelled out here to aid the flow.
4. Page 5, last paragraph of results' first part: 'OSBP bridges...' reference missing.
5. In the results there is no mention of OSBP purification. A sentence on this would help.
6. 'Onions' is sometimes written in French, please correct to English.
7. Figure 2D does not show a 'reconstruction', but a segmentation/rendering. A reconstruction is objective and a rendered one is not.

Reviewer #2 (Remarks to the Author):

Review on the manuscript ,Nanoscale architecture of a VAP-A-OSBP tethering complex at membrane contact site'

Membrane contact sites are important for the communication of organelles in almost every eukaryotic cell. Little is known about the structural properties of membrane contact sites. This study uses elegant, biochemical approaches to study the structural properties of the membrane contact site forming VAP-A protein. By generating a membrane contact site from the bottom up and investigating its structural properties via cryo electron microscopy and FRET, this study represents an important step towards a better understanding how dozens, if not hundreds of proteins, can be accommodated within structurally flexible membrane contact sites.

The study is very well controlled, the technical quality is consistently high, and both the rationale and the importance of the obtained findings are exceptionally well explained. I am not an expert in electron-microscopy, but I can say that applying cryo-EM to such flexible proteins is a challenge. It was mastered by the authors.

Given the importance of membrane contact sites for organelle communication, the potential impact of this study for a broad readership covering the fields from biophysics to cell biology, the technical quality of the data, and the clarity of the presentation, I recommend this manuscript for publication in Nature Communications after addressing the following minor points.

Point 1: The authors make a good job in indicating the importance of the lipid/protein ratio (LPR) throughout the manuscript. However, the authors could provide estimates of the LPR of the given proteins as present in cells. If these numbers are not available for the given proteins, it should be possible to provide numbers/estimates for homologues/analogous proteins in *Saccharomyces cerevisiae*.

Point 2: The authors provide a size distribution of proteoliposomes based on Cryo-EM data. Given that Cryo-EM might 'select' for certain molecular species especially during the freezing process, I would recommend applying also an alternative method (e.g. based on light scattering) to validate the size distribution of the proteoliposomes wherever particularly relevant (reconstitution at 4°C versus at 20°C).

Point 3: The topology of VAP-A after reconstitution is not entirely defined. As stated by the authors, some proteins are reconstituted with the upside up others with the upside down. I do not see this as a major issue for the immediate conclusions drawn by the authors, however, it is possible that under certain special conditions (esp. upon reconstituting VAP-A in smaller liposomes) that the orientation changes substantially. Is it possible to test the orientation e.g. by protease protection or alternative means? I would highly recommend to do so.

Point 4: While I do not think that it would be 100% necessary, it would be great to see FRET data reporting on the distance of the tip of VAP-A from the membrane surface. I am not sure if this is immediately feasible with the constructs at hand, but if possible, I would recommend to do it.

Point 5: In the paragraph 'Formation of membrane contact sites with VAP-A and OSBP' the authors mention that VAP-A was reconstituted in ER-like liposomes. Could they provide more information in the text? Could they provide a rationale for using the given composition?

Point 6: The title should say 'Nanoscale architecture of a VAP-A-OSBP tethering complex at membrane contact site'.

Point 7: There is a typo in the paragraph 'Membrane contact site made with VAP-A can accommodate protein of different sizes'. It says 'We also found VAP-A ribbons forming contact regionsRibbons,'.

Point 8: In Figure 2 the authors indicate structural properties (intermembrane distance) for membrane contact sites as observed via cryoEM. The color coding makes it hard to 'read' the figure. The authors should change to a less arbitrary color coding (e.g. using color-codings as used for AFM data) to make it better readable (also for color-blind individuals).

Point 9: Some electron micrographs still suffer from a relatively poor contrast (especially when printing the figures). Would it be possible to improve the readability in Figure 1 and 3 by either improving contrast or by using labels and larger graphical representations?

Other than that – a beautiful piece of work! Congratulations!

Reviewer #3 (Remarks to the Author):

In this study, the authors designed, generated and characterized in vitro membrane contact sites (MCS), consisting of full-length VAP-A reconstituted into proteoliposomes and OSBP, or a truncated OSBP construct, assembled on lipid nanotubes. A subset of these were sufficiently homogeneous to permit subtomogram averaging that led to 3d models for both VAP-A and part of OSBP.

The authors express VAP-A in *E. coli* and purify it in micellar form, thereafter reconstituting it into PC/PS liposomes, at various lipid:protein ratios, by removal of detergent. Using proteoliposomes reconstituted at a low lipid:VAP-A ratio (high protein density), irregular proteoliposomes were observed, with clear protein density extending from the easily-identifiable lipid bilayer. The extent of the protein layer and its apparent morphology varied depending on the morphology of the host membrane, perhaps reflecting varying arrangements of the VAP-A under these different scenarios. This may be the molecular basis for VAP-A function in forming contacts with many different binding partners and of variable architectures.

Next, the authors use the VAP-A proteoliposomes and lipid nanotubes (containing PI4P to specifically bind OSBP or its truncation construct) to generate a minimal MCS of defined composition. Some optimization of conditions resulted in samples with an appropriate density of MCS to permit their architectural characterization. The detailed architecture of the MCS depended on the VAP-A density in the proteoliposomes, as assessed by measurement of the inter-bilayer distances in the contact sites from 2D images and tomograms and also on whether full length OSBP or its truncation were used, with the presence of the ORD resulting in smaller intermembrane distances.

The authors then perform a more detailed structural characterization of the least heterogeneous contact sites that were observed to form between VAP-A proteoliposomes and nanotubes decorated with truncated OSBP. The least heterogeneous class obtained after 3d classification of the most homogeneous contacts formed were then subjected to characterization by sub-tomogram averaging. Separate models for VAP-A (~20 Å, C1) and NPH-FFAT (~10 Å, C2) were obtained. The NPH-FFAT model was remarkable – a T structure with a long rod parallel to the membrane. Based on modeling and some biochemical validation, the authors propose the long rod parallel to the membrane is due to the region of NPH-FFAT connecting the PH to the GGAT motif (the central core).

The central core was expressed and shown to be dimeric, confirmed by some FRET. MD was used to distinguish between models of the core and the winner fits the envelope quite well.

Overall, this is an interesting study that makes a nice contribution to an extremely challenging structural biology problem. There are a few issues that merit addressing/discussing.

Fig. 1C: A main concern in this study is that the SDS-PAGE analysis of the fractions from the liposome flotation include another band at a size of <70kDa. What is this? Is it some aggregate of VAP-A retaining its Strept tag or a co-incorporating *E. coli* contaminant (or spurious binding partner of VAP-A from detergent-solubilized *E. coli*)? Could this contribute to observed MCS formation? What additional QC was performed on the reconstituted proteoliposomes?

Fig. 1D: Further validation of the experimental system used here relies upon dequenching of the

fluorescence of the FAPP1 PH domain labeled with NBD. The interpretation is that this is due to transfer of PI4P to recipient vesicles containing VAP-A. While OSW-1 clearly inhibits the dequenching, this assay assesses only OSBP function. The authors would be encouraged, at a minimum, to add a control where this experiment is performed using recipient liposomes lacking VAP-A. The structural model for the central core presented later in the manuscript suggests that the central core is capable of dimerizing in the absence of VAP-A so it is at least quite possible that the OSBP tethers and brings the donor and recipient vesicles into close proximity.

It appears that the Central Core is dimeric when expressed in *E. coli* (Fig. 5C). Does the central core of OSBP therefore form T-shaped structures even in the absence of VAP-A or tethering partners? Can it tether without VAP-A?

If the base of the T is not a localized PH domain or pair of PH domains, what is it?

I am struggling with the FRET data as FRET efficiency falls off with the inverse 6th power of distance. Fig. 5I, from the MD, suggests the C-terminus-C-terminus distance is 10 nm while it is <1 nm for the N-termini. The actual FRET signals are not very different at all, certainly not comparable to what might be predicted on the basis of the differing separations.

Small things

Please reread and re-edit the methods – there are numerous typos etc. (particularly FSC0.134).

REVIEWER COMMENTS

Reviewer #1 (Remarks to the Author):

In this manuscript, entitled 'Nanoscale architecture of a VAP-A-OSBP tethering complex at membrane contact sites', de la Mora and colleagues reconstitute membrane contact sites (MCS) in vitro and use cryo-electron tomography to visualise the MCS structure.

The reconstitution is performed by adding a pair of tethering proteins, VAP-A and the OSBP FFAT domain to vesicular and tubular membranes respectively. The authors observe interaction between these two membranes, and this causes deformation of the VAP-A containing vesicles. They observe different types of interactions/deformations, and find that the inter-membrane distance varies between them, suggesting different conformations of the VAP-A protein. Subtomogram averaging is used to derive low-resolution structures of the two protein components and to propose a model of their architecture at MCS. The in vitro reconstitution of MCS is a nice achievement, as well as the observation of heterogeneity that can be explained by VAP-A flexibility. However, I find many of the conclusions beyond this are not well supported by the data.

Specific criticisms are listed below:

Major comments:

1) **Question 1.** Molecular weight and oligomerisation states are derived from gel filtration profiles. SEC is not a reliable methodology to determine molecular weight, and such conclusions should be supported by other methods e.g. SEC/MAIS.

Answer: We agree that an apparent molecular weight as deduced from a size exclusion chromatography, is not a definitive proof of an oligomeric state, especially in the presence of detergent micelles. We have removed the following sentence: “However, the elution peak of VAP-A also preceded

that of BmrA, a membrane protein of 130 kDa solubilized in DDM, suggesting that VAP-A had an oligomeric form larger than a dimer or a non-globular form.”

We have contacted colleagues do perform SEC-MALS on VAP-A. However, the results of Sec were difficult to interpret because the size of the micelle is close to the dimer of VAP-A.

The issue of the presence of dimers in the preparation of purified VAP-A has been also pointed out by reviewer 3, question 21, who noticed the presence of a band between 55 and-70 kDa in the SDS PAGE gel of reconstituted VAP-A in liposomes (Figure 1C). To clarify this point, we have performed additional biochemical characterization of the purified protein. We found that in the presence of reducing agents, VAP-A preparation showed 2 bands at ≈ 55 kDa and ≈ 25 kDa in a SDS PAGE gel, while a single band was found when the samples were heated revealing dimers of VAP-A in the preparation (Figure Q1).

Figure Q1: Purified full length of VAP-A in 12 % SDS PAGE gel in the presence of reducing agents at room temperature or after 5 min incubation at 95°C. The two bands were consistent with dimers and monomers of VAP-A.

Furthermore, VAP-B has been shown to be a dimer (Kim, S. et al., J. Biol. Chem., 2010) and VAP-B sequence shows 63% identity with VAP-A.

Finally, the 3D model of VAP-A derived from our cryo-EM analysis is consistent with a dimer of VAP-A. (see also answer to the Question 7).

2) Question 2. If there is an interaction between OSBP and VAP-A this should be seen by floatation of OSBP when added to the experiments discussed in Figure 1C. Has this test been done? It would further validate that the MCS formation seen by cryo-EM is due to specific protein-protein interaction.

Answer: Full length VAP-A has the same MSP domain as His-VAP-A used in our previous experiments and for which we previously demonstrated an interaction with OSBP or N-PH-FFAT at the surface of liposomes by numerous approaches:

i) Ni-NTA DOGS doped liposomes with bound soluble VAP-A aggregated when mixed with PI4P liposomes and N-PH-FFAT. i) His-VAP-A (8-212) and N-PH-FFAT co-sedimented with PI4P liposomes (Jamecna D., Dev Cell 2019, figure 5A, Mesmin, B., Cell 2013, Figure 2B), iii) His-VAP-A and NPH-FFAT interacted at the interface between 2 Giant Unilamellar Vesicles (Jamecna, D., Dev Cell 2019, Figure 6D). Moreover, we and others have demonstrated that VAP-A specifically binds OSBP at the ER-Golgi in cells (Wyles JP, JBC. 2002, 277:29908-29918, Peresse, J. Biol. Chem. (2020) 295(13) 4277–4288, Fig. S3 A and B, Mesmin, B. Cell 2013, Figure 1).

Notwithstanding, we have performed the experiment suggested by the reviewer; i.e. flotation of N-PH-FFAT with full length VAP-A reconstituted in proteoliposomes (Figure Q2). Experimental conditions were the same as sedimentation described Figure 1C except that N-PH-FFAT was incubated with VAP-A proteoliposomes. After sedimentation, N-PH-FFAT was found associated with a single band also containing VAP-A demonstrating that NPH-FFAT binds to VAP-A full length (fractions 7-9).

Figure Q2-a: Binding of N-PH-FFAT to VAP-A. VAP-A proteoliposomes reconstituted at LPR 1400 mol/mol in PC/PS lipid mixture were mixed with N-PH-FFAT at 1/1 molar ratio before sedimentation in a 30/20/10/5% discontinuous sucrose gradient. After overnight centrifugation at 37 000 rpm, in a TLS 55 swing rotor, fractions were collected and analyzed by SDS PAGE gel.

We also present additional indications of VAP-A / NPH-FFAT interaction by cryo-EM that we did not show in the manuscript due to the lack of space.

Figure Q2-b shows that i) Pure PC/PS lipid vesicles used for reconstitution of VAP-A did not bind to N-PH-FFAT Pi4P Gal tubes in a non-specific manner (Figure Q2b, A-B), ii) Small proteoliposomes of VAP-A full length reconstituted at 4°C, as depicted in Sup Fig 3D, bind PI4P-gal tubes in the presence of N-PH-FFAT (Figure Q2b-D) but not in the absence of N-PH-FFAT (Figure Q2b -C), iii) Soluble human VAP-A (8-212) did not bind to Pi4P Gal tubes in the absence of NPH-FFAT (Figure Q2b-E).

Altogether, these experiments demonstrate that full length VAP-A specifically binds to N-PH-FFAT and form MCSs.

Figure Q2b: Control experiments of VAP-A proteoliposomes and N-PH-FFAT interactions. (A, B) 100 μM Pure PC/PS lipid vesicles were incubated with 50 μM PI4P 30-Gal tubes in the presence of 30 μM N-PH-

FAT. Tubes were covered by N-PH-FFAT (open arrows) but no vesicles were bound to tubes (black arrows). (C) 100 μ M (2.5 μ M VAP-A) small VAP-A proteoliposomes did not bind to PI4P Gal tubes in the absence of N-PH-FFAT but did so (D) in the presence of N-PH-FFAT. (E) 2.5 μ M soluble VAP-A-6-His (8-212) was incubated with 50 μ M PI4P Gal tubes but did not bind to them. Scale bars in (A, C): 500 nm, in (D) 50 nm, in (B, E): 50 nm.

3) **Question 3.** On page 7, the authors discuss ‘dark dots’ and assign them to specific domains. There is no reason to think that certain domains would be darker than others and this discussion should be removed.

Answer: We respectfully disagree with the reviewer. Models and structural studies of VAP-A suggest a long coiled-coil region followed by a folded domain, MSP, at the tip (Reviewed in Murphy S and Levine T, BBA 2016, Figure 1, Kaiser A et al., Structure 2005, Figure 4). Furthermore, our 3D model of VAP-A depicted figure 5D is consistent with the MSP domain at the tip of the molecule.

To add more supports on this assumption, we have purified a construct with MSP domain only, VAP-A [8-139]-6 \times His, and imaged it bound to Ni-NTA lipid tubes. As show below, small electronic densities were found on the tubes that were similar to those reported at the tip of full length VAP-A.

Figure Q3: Cryo-EM images of His-MSP (A) bound to Ni-NTA-DOGS gal tubes. MSP (8-139) domains appear in (A) as dots that are also found in Figure 1 F. Bars: 20 nm

4) **Question 4.** On page 7, the authors discuss that the elongated shape is consistent with high molecular weight size-exclusion chromatography. See point 1, this should be removed.

Answer: Agree. We have removed this sentence.

5) **Question 5.** In Figure 2, the authors suggest that the full-length OSBP imposes shorter space between opposing membranes because it is less packed and therefore VAP-A adopts an elongated conformation. Given that difference distances (attributed to different VAP-A conformations) are also seen in round vesicles versus flattened ones versus ribbons, this means there should be a correlation. I.e. MCS reconstituted with full-length OSBP should be enriched in vesicles over ribbons, and viceversa? Do the authors observe this and if not, why?

Answer: We thank the reviewer for this excellent suggestion, which prompted us to further analyze our reconstitutions. As predicted by the reviewer, we observed more spherical or slightly deformed VAP-A vesicles in contact with OSBP tubes than with N-PH-FFAT tubes. For this aim, we analyzed 45 OSBP tomograms and the 94 NPHFFAT tomograms. We observed:

42 spherical vesicles and 47 flat vesicles for the 89 contacts induced by OSBP, i.e. 48% and 52%, respectively

67 spherical vesicles and 158 flat vesicles among the 226 contacts induced by N-PH-FFAT, i.e. 30%.

In addition, we found 17 contacts between ribbons and tubes for N-PH-FFAT, but none in the case of OSBP. This observation indicates that when VAP-A is too packed as in ribbons, OSBP cannot form contact despite the presence of the FFAT sequence.

We have added this information in the text

Spherical VAP-A vesicles, in which VAP-A is present at low density, were involved in 48% of the MCSs induced by OSBP (n=89) as compared to 30% in the case of the MCSs induced by N-PH-FFAT (n=226). In contrast, no MCS was found between OSBP and the VAP-A ribbons, i.e. the membrane in which VAP-A is present at the highest density.

6. Question 6. How is the classification shown in Figure 4B performed? This is missing from the method and it is not clear whether it is a bona fide classification, or whether the data was simply 'divided' in groups depending on distance measured? Why 4 classes? These details are significant and must be included.

Answer: We apologize for the lack of information on this procedure. The classification is a bona fide process. We have added the sentences below:

We performed a Multireference Analysis (MRA) in Dynamo, using as initial seeds averages of four random combinations of particles; classes representing different intermembrane lengths arise spontaneously. The iterative process stopped after no swapping of particles between classes was observed.

The class number was deemed adequate for a proper balance between the number of particles per class and capturing the main feature of the heterogeneity in the data.

7. Question 7. How was the threshold established for the VAP-A structure in figure 4D? The authors choose an (arbitrary as far as I can see) threshold, fit a dimer, and derive information about the oligomeric state of the protein. They could have equally fit a dimer and assigned a threshold to match its density. Given the absence of recognisable features I question the reliability of size measurements of VAP-A.

We first answered to question 8 and next question 7.

Answer: Threshold decision is indeed a step where a certain subjectivity is introduced in the analysis. In spite of recent attempts for introducing unbiased criteria (as the False Discovery Rate based method in Beckers et al, IUCrJ, 2020 or the automation oriented Pfab et al. in Proceedings of BCB'19), the task of establishing a totally objective threshold remains unresolved, especially for low resolution and membrane proteins.

A classical approach for this task relies on the visual inspection of the behavior of the resulting isosurface for different threshold levels. In the case of the N-PH-FFAT in Figure 4E we have worked with the density in panel a) from the Figures Q6 below as it corresponds to a point where the intensity distribution in the map showed a change of derivative.

a)

b)

Thresholds slightly above this point induced a disappearance of the membrane connection (top row in the following Figure), while thresholds below this point result in saturating the picture (bottom row):

An important point that reassures us about the validity of our choice and the interpretation derived from it is that, even when decreasing the threshold level beyond interpretability of the membrane area and well into the domain where noisy regions became visible, the mean features of the protein remained stable:

Concerning VAP-A, we agree with the referee that it is difficult to discern between a monomer and a dimer at the obtained map resolution of VAP-A (19.5 Å). However, the threshold wasn't arbitrary selected. As explained above, the threshold was determined by identifying the inflection point in the map histogram calculated by Chimera:

Below this point the density was not continuous,

while above the selected threshold value, noise appeared:

The dimer was selected over the monomer based on the correlation calculated by Chimera, 0.93 for the dimer vs 0.89 for the monomer, upon fitting the model into the map by simulating a map/calculating a simulated map at 20 Å. The lower correlation was related to the lower mass of the monomer compared to that of the calculated map. A tetramer was also fitted into the map but its volume surpassed that of the calculated map. The estimated correlation was 0.80 in this case.

8. Question 8. Similarly, for the N-PH-FFAT density in Figure 4E: how was the threshold decided?

Idem Q7)

Answer: see above.

9. Question 9. For both structures, I have concerns over their validation. I am sorry to say that the density of N-PH-FFAT does not appear to reach a resolution anywhere close the claimed 9.8Å. While I commend the inclusion of supplementary Figure 4, I am concerned about noise alignment, seeing how the density abruptly disappears outside the masks used. It is not clear in the methods at which point the dataset was split in two halves: before alignment or only prior to FSC calculation?

The reviewer is expressing two points of concern: first that the reported resolution does not correspond to the visual appearance of the map; second that the map might contain some degree of template bias.

First point: the result we wished to present was the density itself, rather than the numerical resolution. We agree that, at the end of the day, a density is as good as its visual appearance. We provide the resolution attained with Relion when its postprocessing module was fed with our halfmaps and the mask used during iterative alignment; the reported resolution was the overall structure provided by that software.

According to theory (Chen et al, JSB 2013), phase randomization should provide an objective resolution estimate that does not depend on the used mask; the reality is that at the resolution levels attained in this study, different masks provide different curves. Here we paste two different resolution curves obtained for the same halfmaps, both using smooth masks: the left one is a mask with the shape of the protein, the right one is a spherical mask (Figure Q9). Following the output of the post-processing module of Relion, the FSC obtained with phase randomization is depicted in red; green is the FSC of unmasked maps, black and blue reflect the overestimation of resolution stemming from including the mask in the FSC computation.

Prompted by the reviewer's observation, we have further investigated the map of local resolutions as detected by the blocres function in the Bsoft package (Heyman, JB, Protein Sc. 2021). The result shows that densities range between 8 and 19 Å, with the bulk of the structure in the range of 12-14 Å. This depicts a more realistic representation on the visual appearance of the average that the single value delivered by Relion.

Accordingly, we have changed the sentence:

<<The resolution for VAP-A and of NPH-FFAT as determined by the FSC0.134 was 19.6 Å and 9.8 Å, respectively>>

by

The overall resolutions for VAP-A and of NPH-FFAT as determined by the FSC0.134 was 19.6 Å and 9.8 Å, respectively. For NPH-FFAT, a further inspection of the local resolution computed by the program blocres (BSoft) pointed at a distribution of resolutions in the range of 9-18Å.

About the possibility of noise alignment, we have followed a strict golden-standard type of computation, splitting the two halves of the data set before iterative alignment.

10) **Question 10.** Is the elongated shape of N-PH-FFAT also seen in the raw data? How do the subunits orient with respect to each other and to the membrane in 3D? Have the authors checked that these long rods do not clash with each other, or the VAP-A with the N-PH-FFAT densities?

Answer. The elongated shape of N-PH-FFAT was not observed in the raw data. In raw images and in reconstituted images we were able to see the proteins bound to PI4P containing tubes (for some examples see Figure 2B of the original submitted manuscript) but we cannot have an insight into the elongated shape. This shape is observed upon several alignment steps of the extracted subvolumes as described in the text of SuppFigure 4. However, see also answer Question Q23.1, we have performed a 2D analysis of N-PH-FFAT bound to PI4P tubes and found a density at 5 nm from the membrane and tiny densities that join the outer lipidic leaflet and this density. These features are consistent with the T shape of central core OSBP.

By plotting the position and orientation of the aligned subvolumes we confirm that the subunits are oriented orthogonal to the tube axis. As an example, figure Q10 below shows the distribution of the aligned N-PH-FFAT subvolumes with their respective 'z' axis in blue and their 'x' axis in red. The 'x' axis is perpendicular to the elongated shape of the average shown in Figure 4E of the submitted version of our manuscript. The 'x' axis is randomly oriented. There is no clash between N-PH-FFAT subvolumes because the closest distance between them (8.5 nm) is larger than half of the dimer (6 nm).

Figure Q10.1. Localization and orientation of the aligned subvolumes that upon averaging gave the map at 9.8 Å shown in Figure 4E. The blue squares represent the base (part that interacts with the membrane bilayer) of each sub-volume.

Following the same approach, and representing the final sub-volumes corresponding to VAP-A (green) and N-PH-FFAT (blue) we can demonstrate that there is no clash in the contact zone between the sub-volumes corresponding to VAP-A and those corresponding to N-PH-FFAT, as observed in the figure below:

Figure Q10.2. Localization and orientation of the aligned subvolumes that upon averaging gave the map at 11.05 Å shown in Figure 4E. The blue squares represent the base of N-PH-FFAT and the green square the MSP density of VAP-A.

The organization of the contact sites is that described on Figure 4c (specifically class 3) of the original submitted manuscript): N-PH-FFAT points at ~ 6nm from the tube's membrane, while VAP-A is located at ~ 12 nm of the same membrane.

11) **Question 11.** Finally, I also have concerns regarding the reliability of the model fitted in the N-PH-FFAT density. Firstly, the authors assume the ordered part of the structure is composed by two helices based on secondary structure predictions. Secondly, they perform FRET experiments to establish N-termini are in close proximity (assuming dimerization, which is only suggested by SEC analysis). Models are next built with Robetta, and chosen based on stability as assessed by Molecular Dynamics. All in all, this process makes a number of assumptions that are not experimentally validated.

Answer: We respectfully disagree with the reviewer. We have previously demonstrated the dimeric form of NPH-FFAT by size-exclusion chromatography but also by proteolytic experiments (Mesmin, B., Cell 2013). We also have demonstrated the dimeric form of OSBP in experiments of binding to Pi4P liposomes and the measurement of Kon/Koff in comparison with a monomeric construct of OSBP lacking the Central Core region (Jamecna, D, Dev Cell 2019).

Here, we present circular dichroism data that show that the central core of OSBP is mostly α -helical. The FRET experiments show a signal between N-termini labelled proteins; obviously this requires a species with at least two monomers. The model of dimer of Central core OSBP derived from MD is stable along 500 nsec while other models of dimers made from monomers like the one depicted in Figure S5 were unstable after only 50 nsec.

We have now added a new FRET experiment that confirmed that N-termini and C-termini are close and distant in the dimer, respectively (Figure 5 revised) (see also answer to question 25).

Minor comments:

1. References are missing throughout the first part of the introduction

We are not sure which references were missing and we have added 8 new references in the Introduction (in red in the text). There are now 68 references among 50 suggested for an Article in Nature communications.

2. Page 4, 1st sentence: “this contact site”. Which?

Corrected

3. Same sentence as above. While it becomes clear later that OSBP contains FFAT motif, this should be spelled out here to aid the flow.

Corrected

4. Page 5, last paragraph of results’ first part: ‘OSBP bridges...’ reference missing.

A reference has been added

5. In the results there is no mention of OSBP purification. A sentence on this would help.

The following sentence has been added:

N-PH-FFAT and OSBP were purified as reported²³.

6. ‘Onions’ is sometimes written in French, please correct to English.

Corrected

7. Figure 2D does not show a 'reconstruction', but a segmentation/rendering. A reconstruction is objective and a rendered one is not.

Corrected, thank you.

Reviewer #2 (Remarks to the Author):

Review on the manuscript 'Nanoscale architecture of a VAP-A-OSBP tethering complex at membrane contact site'.

Membrane contact sites are important for the communication of organelles in almost every eukaryotic cell. Little is known about the structural properties of membrane contact sites. This study uses elegant, biochemical approaches to study the structural properties of the membrane contact site forming VAP A protein. By generating a membrane contact site from the bottom up and investigating its structural properties via cryo electron microscopy and FRET, this study represents an important step towards a better understanding how dozens, if not hundreds of proteins, can be accommodated within structurally flexible membrane contact sites. The study is very well controlled, the technical quality is consistently high, and both the rationale and the importance of the obtained findings are exceptionally well explained. I am not an expert in electron-microscopy, but I can say that applying cryo-EM to such flexible proteins is a challenge. It was mastered by the authors.

Given the importance of membrane contact sites for organelle communication, the potential impact of this study for a broad readership covering the fields from biophysics to cell biology, the technical quality of the data, and the clarity of the presentation, I recommend this manuscript for publication in Nature Communications after addressing the following minor points.

Question 12. Point 1: The authors make a good job in indicating the importance of the lipid/protein ratio (LPR) throughout the manuscript. However, the authors could provide estimates of the LPR of the given proteins as present in cells. If these numbers are not available for the given proteins, it should be possible to provide numbers/estimates for homologues/analogous proteins in *Saccharomyces cerevisiae*.

Answer: Thank you for this interesting question. As requested by the reviewer, we have compared the VAP lipid/protein ratio that we used in in vitro experiments to the one that can be estimated in cells. We noted that this estimate would only make sense in the context of a membrane contact sites. However, we note that there are few data on the concentration of proteins in MCSs. Nevertheless, we made the following estimate:

- The VAP (VAPA+B) copy-number/cell is $\approx 0.6 \times 10^6$ (Hein MY et al, Cell, 2015).

- There are about 10^9 lipid molecules in the plasma membrane in a mammalian cell (*Molecular Biology of The Cell*, 4th edition. Alberts B, Johnson A, Lewis J, et al. 2002). However, it is assumed that the PM contains only 5% of the total lipids of the cell, against 60% for the ER because of its large surface area (*Cell Biology by the numbers, First Edition, Milo R & Philips R. 2015, Garland Science*). We can therefore estimate that ER has 12×10^9 lipids. The lipid/(VAPs) would then be 20,000 mol/mol. However, fluorescence imaging observations by our group and by others indicate that VAP and OSBP show a ~10-fold enrichment at ER/TGN MCSs relative to the rest of the cell. This situation may be further amplified when cells are treated with an OSBP inhibitor such as OSW-1. In this context, an extra 10-fold increase in concentration can be estimated, suggesting 200 lipids/VAP. A LPR=200 mol/mol is typically in the range of our in vitro reconstituted MCSs. This reinforces the idea that MCSs are confined regions with protein complexes at high densities and in close interaction.

Question 13. Point 2: The authors provide a size distribution of proteoliposomes based on Cryo-EM data. Given that Cryo-EM might 'select' for certain molecular species especially during the freezing process, I would recommend applying also an alternative method (e.g. based on light scattering) to validate the size distribution of the proteoliposomes wherever particularly relevant (reconstitution at 4°C versus at 20°C).

Answer: We agree that cryo-EM grids can select under some conditions liposomes depending on their size and shape. This is the case for Quantifoil or C-flat grids with calibrated holes, that are often use for single particle analysis of proteins, and when liposomes are blotted from the drop side. However, we always used lacey grids with a networks of holes of size ranging from 50 nm to several microns. In addition, we blotted the solution of liposomes from the back side forcing the liposomes to go through holes, as also reported in studies of membrane machineries (Kovtun, O, *Nature* 2018, Dodonova, *Elife* 2017, Zanetti, G., *Elife* 2013). This also enabled to decrease the concentration of liposomes by 10-50 fold to 50 μ M as compared to > 0.5 mM in Quantifoil grids. In the case of the formation of VAP-A and NPH-FFAT MCSs, it was crucial to use low concentration of proteoliposomes and Galtubes; otherwise it led to massive aggregation onto the grid.

We have performed the comparison of two types of grids and blotting condition. We used a solution of proteoliposomes of VAP-A reconstituted at LPR 70 mol/mol since it contains liposomes that are variable in shape and size (**Figure Q13**). In **Figure Q13.A**, with lacey grids, as depicted in Figure 1 of the manuscript, we found large and flat vesicles, onions, small and large spherical vesicles and ribbons. This was obtained with 50 μ M of proteoliposomes and back side blotting. In **Figure Q13.B**, with Quantifoil R1/R2, no material or just a few (arrow) were found at the same proteoliposomes concentration. In **Figure Q13.C**, we increased the concentration of proteoliposomes to 0.5 mM. Proteoliposomes were almost found on the carbon support (with grid washed or not with CHCL3) while only small vesicles were found in the holes.

We also did DLS measurement as suggested by the reviewer. We reconstituted VAP-A in proteoliposomes at LPR 70 mol/mol at RT or at 4°C. For the reconstitution at RT, we measured an averaged radius of 90 nm (95% of mass) and $r = 234$ nm (5% of mass) (**Figure Q13.D**). By cryo-EM, we observed for the reconstitution at RT, a more heterogeneous population of vesicles than by DLS with vesicles ranging from 50 nm to 1-2 microns. We cannot rule out that larger vesicles were entrapped and enriched in the holes. For the reconstitution at 4°C, we measured a homogeneous population of $r = 18.64$ nm (89% of mass) and two minor populations ($R = 72.7$ nm, 8.7 % of mass, 222.89 nm and 1.4 % of mass) (**Figure Q13.E**). For reconstitution at 4°C, reported sizes of proteoliposomes from cryo-EM in Figure 3D were of $R = 150$ nm \pm 5 nm, consistent with DLS. Finally, it is worth noting that the

intermembrane distances reported along the manuscript were measured on images where individual vesicles, with their size and shape, in contact to GaITubes were clearly identifiable.

Figure Q13: (A-C) Comparison of cryo-EM grids for imaging VAP-A proteoliposomes. VAP-A proteoliposomes reconstituted at LPR 70 mol/mol at RT were deposited at 0.05 mg/ml on lacey grids (A), 0.05 mg/ml (B) or 0.5 mg/ml (C) on Quantifoil grids. Samples were blotted from the back side in (A) or on the drop side (B, C). The various populations of proteoliposomes are indicated: onions (white arrows), angular shape liposomes (dark arrows), ribbons (empty dark arrows), spherical liposomes (empty white arrows).

Question 14. Point 3: The topology of VAP-A after reconstitution is not entirely defined. As stated by

the authors, some proteins are reconstituted with the upside up others with the upside down. I do not see this as a major issue for the immediate conclusions drawn by the authors, however, it is possible that under certain special conditions (esp. upon reconstituting VAP-A in smaller liposomes) that the orientation changes substantially. Is it possible to test the orientation e.g. by protease protection or alternative means? I would highly recommend to do so.

Answer: We thank the reviewer for this remark.

In in vitro MCSs, only VAP-A oriented with the MSP domains pointing outward, i.e. MSP-out orientation, were involved in the formation of MCSs. VAP-A in the opposite orientation, i.e. MSP-in, did not participate to the formation of MCSs.

We analyzed two types of reconstitutions: - reconstitutions made at RT leading to a population of vesicles, ribbons, multilayered vesicles, - reconstitution at 4°C with homogenous population of small liposomes;

We have evaluated the orientation of VAP-A through two approaches: - Biochemical measurement of the orientation of VAP-A in the population of reconstituted VAP-A.- Direct visualization of VAP-A orientations on cryo-EM images of proteoliposomes forming MCSs with N-PH-FFAT or OSBP.

For the biochemical measurement, the principle of the assay relied on adding trypsin to proteoliposomes to proteolyze only MSP-out VAP-A and not MSP-in VAP-A, proteoliposomes being not permeable to trypsin. Thereafter, we compared the amount of digested VAP-A in proteoliposomes with digested VAP-A on solubilized proteoliposomes with Triton X100 (complete digestion of VAP-A) and with proteoliposomes not treated with trypsin (100% VAP-A) (Figure Q14.A). From the results we found that VAP-A was oriented 40/60 out/in on average after reconstitution at RT and 50/50 after reconstitution at 4°C in small liposomes. The presence of multilayer vesicles in reconstitution at RT, where most VAP-A should not be not accessible to trypsin can explain the lower value of trypsin sensitive VAP-A. Overall, this assay show that VAP-A molecules are symmetrically inserted in the bilayer.

For the analysis of cryo-EM images.

Reconstitution at RT: - 1) In the case of flat VAP-A proteoliposomes forming MCSs (figure 2D), we observed on cryo-tomograms that VAP-A was incorporated in both orientations (Figure Q14.B, C, D). 2) In the case of VAP-A in ribbons, we found images where VAP-A was also oriented symmetrically with the lipid bilayer in the middle. 3) In the case of large and spherical vesicles, we did not see VAP-A molecules but we showed biochemically that VAP-A was incorporated in both orientations. Thus, whatever the VAP-A concentration in the membrane, only 50% of VAP-A were involved in the formation of MCS. This does not change our conclusion that intermembrane distances increase with the concentration of VAP-A in MCSs.

Reconstitution at 4°C.: We showed that small proteoliposomes e.g. at LPR 1400 mol/mol, formed MCSs with 15 nm short intermembrane distances revealing that few VAP-A molecules were involved in the MCSs (Figure 3D). When protein concentration was increased 20 fold from 1400 to 70 lipid/protein ratio, the intermembrane distances did not change, only few VAP-A were still involved in the formation of MCSs. (Figure 3D). It is thus likely that if some VAP-A were oriented MSP-in, it would not modify the local concentration of VAP-A in MCSs.

Figure Q14: Orientation of VAP-A in proteoliposomes in contact with N-PH-FFAT or OSBP. A) Orientation of VAP-A in proteoliposomes reconstituted at RT and at 4°C. Proteoliposomes were not treated (lane 0), digested at the indicated time incubation with trypsin at a VAP-A/Trypsin 100 or 50 mol/mol before or after (lanes Triton) solubilization with Triton X100 (Triton X100/lipid at 2.5 w:w). (B) representative section of a cryo-tomogram of two VAP-A flat vesicles forming MCSs with N-PH-FFAT. VAP-A oriented with MSP pointing toward the interior of the vesicles are visible (red arrows) showing that VAP-A is oriented 50/50 in the membrane. (C) small vesicles of VAP-A in contact with N-PH-FFAT (light green arrow) or not (dark green arrow). VAP-A were not visible due to their low concentration in the membrane. (D) ribbons of highly packed VAP-A oriented 50/50 in the central lipid bilayer. (E) Scheme illustrating how VAP-A is oriented in proteoliposomes reported in this study. Only MSP-out VAP-A are

involved in the formation of MSC. The intermembrane distances increase with the concentration of MSP-out VAP-A in the membrane.

In the manuscript, we have added:

VAP-A was oriented symmetrically in the membrane of vesicles or ribbons, as shown in cryo-tomograms (see below) but we considered that only VAP-A pointing outward vesicles were involved in the formation of MCSs.

Question 15. Point 4: While I do not think that it would be 100% necessary, it would be great to see FRET data reporting on the distance of the tip of VAP-A from the membrane surface. I am not sure if this is immediately feasible with the constructs at hand, but if possible, I would recommend to do it.

Answer: We appreciated the reviewer's idea, and as suggested we conducted FRET experiments. For this, we used soluble VAP-A-6His (8-212) bound to liposomes containing DGS-NTA-Ni lipids. These liposomes also contained Rhodamine-PE lipids (as FRET acceptor). We then added an AlexaFluor488-labelled peptide labelled on the Cyst residue (WCSGKGDMSDEDDENEFFDAPEIITMPENLGH) comprising a FFAT motif to bind VAP-A (Mesmin, B., Cell 2013). The labelled FFAT peptide was used as FRET donor. If VAP was flexible enough, FRET might occur between the FFAT peptide and the membrane. However, we did not observe any FRET signal. As a control of our experiment, we designed and purified a shorter VAP construct consisting only of its MSP domain (MSP-His). Unfortunately, even with this short construct bound to liposomes, we did not obtain any FRET signal. It is possible that the peptide is too long and that the distance between donor and acceptor is too far. An alternative possibility is that the interaction between the FFAT motif and the MSP is too weak (reported $K_d > 1 \mu M$) and the FRET signal is confounded by the large fraction of free FFAT peptide.

Question 16. Point 5: In the paragraph 'Formation of membrane contact sites with VAP-A and OSBP' the authors mention that VAP-A was reconstituted in ER-like liposomes. Could they provide more information in the text? Could they provide a rationale for using the given composition?

Answer: We apologize for this over simplification. The lipid composition of the ER membrane is more complex. Since no lipid specificity has been reported for the structural organization or function of VAP-A, the liposomes that we used contains 95% ePC as background lipid and 5% PS, a content typical of ER membranes. We used PC and PS from natural sources as they contain a mixture of acyl chains of variable lengths and saturation to ensure a fluid state of the membrane. In PI4P transport experiments, we doped the lipid mixture with 10% cholesterol. We now give more details about the lipid composition in the Material and Methods and remove the term "ER-like" when dealing with MCSs.

Question 17. Point 6: The title should say 'Nanoscale architecture of a VAP-A-OSBP tethering complex at membrane contact site"S"'.
S"

Thanks, corrected.

Question 18. Point 7: There is a typo in the paragraph 'Membrane contact site made with VAP-A can accomodate protein of different sizes'. It says We also found VAP-A ribbons forming contact regionsRibbons,'.

Corrected.

Question 19. Point 8: In Figure 2 the authors indicate structural properties (intermembrane distance) for membrane contact sites as observed via cryoEM. The color coding makes it hard to 'read' the figure. The authors should change to a less arbitrary color coding (e.g. using color-codings as used for AFM data) to make it better readable (also for color-blind individuals).

Answer: This has been corrected in the new Figure 2.

Question 20. Point 9: Some electron micrographs still suffer from a relatively poor contrast (especially when printing the figures). Would it be possible to improve the readability in Figure 1 and 3 by either improving contrast or by using labels and larger graphical representations?

Answer: Cryo-EM images are intrinsically at low contrast. We are not favorable to artificially increase the contrast on raw data. A way should have been to bin pixels but at the expense of resolution. The movie S4 related to Figure 3 shows the spreading of vesicles along the tube and the tongue with a better contrast than the 2D image. It is also worth noting that the size of the images and thus the resolution, was restricted for the initial submission to Nature Communications. We hope that if the manuscript is accepted, the new figures at higher resolution will be more readable.

Other than that – a beautiful piece of work! Congratulations!

Thank you

Reviewer #3 (Remarks to the Author):

In this study, the authors designed, generated and characterized in vitro membrane contact sites (MCS), consisting of full-length VAP-A reconstituted into proteoliposomes and OSBP, or a truncated OSBP construct, assembled on lipid nanotubes. A subset of these were sufficiently homogeneous to permit subtomogram averaging that led to 3d models for both VAP-A and part of OSBP.

The authors express VAP-A in *E. coli* and purify it in micellar form, thereafter reconstituting it into PC/PS liposomes, at various lipid:protein ratios, by removal of detergent. Using proteoliposomes reconstituted at a low lipid:VAP-A ratio (high protein density), irregular proteoliposomes were observed, with clear protein density extending from the easily-identifiable lipid bilayer. The extent of the protein layer and its apparent morphology varied depending on the morphology of the host membrane, perhaps reflecting varying arrangements of the VAP-A under these different scenarios. This may be the molecular basis for VAP-A function in forming contacts with many different binding partners and of variable architectures.

Next, the authors use the VAP-A proteoliposomes and lipid nanotubes (containing PI4P to specifically bind OSBP or its truncation construct) to generate a minimal MCS of defined composition. Some optimization of conditions resulted in samples with an appropriate density of MCS to permit their architectural characterization. The detailed architecture of the MCS depended on the VAP-A density in the proteoliposomes, as assessed by measurement of the inter-bilayer distances in the contact sites

from 2D images and tomograms and also on whether full length OSBP or its truncation were used, with the presence of the ORD resulting in smaller intermembrane distances.

The authors then perform a more detailed structural characterization of the least heterogeneous contact sites that were observed to form between VAP-A proteoliposomes and nanotubes decorated with truncated OSBP. The least heterogeneous class obtained after 3d classification of the most homogeneous contacts formed were then subjected to characterization by sub-tomogram averaging. Separate models for VAP-A (~20 Å, C1) and NPH-FFAT (~10 Å, C2) were obtained. The NPH-FFAT model was remarkable – a T structure with a long rod parallel to the membrane. Based on modeling and some biochemical validation, the authors propose the long rod parallel to the membrane is due to the region of NPH-FFAT connecting the PH to the GGAT motif (the central core).

The central core was expressed and shown to be dimeric, confirmed by some FRET. MD was used to distinguish between models of the core and the winner fits the envelope quite well.

Overall, this is an interesting study that makes a nice contribution to an extremely challenging structural biology problem. There are a few issues that merit addressing/discussing.

Question 21) Fig. 1C: A main concern in this study is that the SDS-PAGE analysis of the fractions from the liposome flotation include another band at a size of <70kDa. What is this? Is it some aggregate of VAP-A retaining its Strep tag or a co-incorporating E. coli contaminant (or spurious binding partner of VAP-A from detergent-solubilized E. coli)? Could this contribute to observed MCS formation? What additional QC was performed on the reconstituted proteoliposomes?

Answer: The 60 kDa band present in the gel from the liposome flotation is not a contaminant but dimeric VAP-A. In SDS PAGE gel, purified VAP-A denatured in the presence of the reducing agents B-mercapto-ethanol or TCEP and heated at 95° showed a single band without any contaminant (Figure Q1). We used such preparation of proteins for the sedimentation of liposomes and thus we did not expect new bands. However, due to the presence of sucrose in the fractions, we did not heat the samples at 95°C and in such conditions, as shown in the Figure 1D, VAP-A runs as two bands at 60 kDa and 30 kDa for the dimeric and monomeric state, respectively (Figure 1R).

Figure 1R: Purified full length of VAP-A in 12 % SDS PAGE gel in the presence of reducing agents at RT or after 5 minutes incubation at 95°C. Molecular weights were consistent with dimers and monomers of VAP-A.

We have added in the Figure legend 1 the following sentence:

Fractions from the sucrose gradient were not heated at 95°C as done in 1B in the denaturing buffer explaining the presence of the bands at 60 kDa corresponding to non-dissociated dimer of VAP-A

Question 22) Fig. 1D: Further validation of the experimental system used here relies upon dequenching of the fluorescence of the FAPP1 PH domain labeled with NBD. The interpretation is that this is due to transfer of PI4P to recipient vesicles containing VAP-A. While OSW-1 clearly inhibits the dequenching, this assay assesses only OSBP function. The authors would be encouraged, at a minimum, to add a control where this experiment is performed using recipient liposomes lacking VAP-A. The structural model for the central core presented later in the manuscript suggests that the central core is capable of dimerizing in the absence of VAP-A so it is at least quite possible that the OSBP tethers and brings the donor and recipient vesicles into close proximity.

Answer: We have previously shown that OSBP is functional only when it is bound to VAP-A, forming a tethering complex between ER and Golgi like membranes (Mesmin B, Cell 2013). This is different from the lipid transport cycle of OSH4 that does not require VAP-A to shuttle lipids between ER and Golgi like liposomes. In the absence of VAP-A, OSBP is unable to transport PI4P (Mesmin B, Cell 2013, Figure 6A, 7E) or to transport cholesterol (Mesmin B, Cell 2013, Figure 5C).

In addition, we have shown that a long 90 aa intrinsically unfolded sequence at the N terminus of OSBP controls its orientation and dynamics at MCS. This prevents the two PH domains of the OSBP dimer from homotypically tethering two Golgi-like membranes (Jamecna D., Dev Cell 2019). Our assays were performed with full length OSBP. We have added the following sentence in the text describing the PI4P-transfer assay:

Previous biochemical reconstitutions and cellular observations indicated that interaction with membrane-bound VAP-A determines the lipid exchange activity of OSBP

Question 23) It appears that the Central Core is dimeric when expressed in *E. coli* (Fig. 5C). Does the central core of OSBP therefore form T-shaped structures even in the absence of VAP-A or tethering partners? Can it tether without VAP-A?

Answer: The Central Core has a size of 140 x 2 amino acids (~ 32 kDa), which is too low for cryo-EM and single particle analysis. It is thus difficult to ambiguously answer to this question.

However, we have performed a 2D images analysis of N-PH-FFAT, i.e. Central Core OSBP and PH domains bound to PI4P tubes. Images have been recorded with our medium resolution cryo-electron microscope (Lab6, 200 kV, F416 TVIPS camera) and analyzed with Xmipp package.

2D classes of averaged protein densities revealed three lines of densities. The first two lines separated by 4-5 nm correspond to the lipid leaflet of the PI4P gal tubes (red circles). In some classes, the outer leaflet is dotted, possibly due to PH domains that have been averaged. A third line is at 5 nm from the outer leaflet and corresponds to the central core OSBP parallel to the membrane (yellow circles). In some case, we see tiny densities that join the outer lipid leaflet and the central core OSBP and may correspond to the stem. This suggests that the T-shaped structure is also present in the absence of bound VAP-A.

Figure Q23.1. Image analysis of N-PH-FFAT bound to PI4P Gal tube in the absence of VAP-A. (A) Cryo-EM image have been recorded with a Lab6 200 kV Tecnai electron microscope and a F416 TVIPS camera at 2.13 Å/px. 900 boxes of 100x100 px were extracted and submitted to 2D classifications using XMipp software. (B) Gallery of the 40 mains 2D classes. Boxes dimensions are 20x 20 nm. Three lines of densities are often present with the two lipidic leaflet (red circles) and a third density at 5 nm that could correspond to the central core OSBP (yellow circles).

As stated above, the central core OSBP does not contain a PH domain to bind Pi4P and thus cannot tether two membranes. We confirmed by cryo-EM that Central Core OSBP did not bind to Pi4P tubes.

Question 24) If the base of the T is not a localized PH domain or pair of PH domains, what is it?

Answer: We observed densities in which a PH domain can be fitted, but as shown in Figure R the resolution is low in this region. We thus prefer to not propose an assignment for the stem. However, we modified the sentence in the discussion by adding the possibility that part of the PH domains constitutes the stem of the T.

The tethering region of OSBP is organized as a T-shaped with a 14 nm elongated domain parallel to the membrane and a short ≈ 3 nm stem, which probably follows or partly contains the two PH domains, connects this elongated structure to the membrane.

Question 25) I am struggling with the FRET data as FRET efficiency falls off with the inverse 6th power of distance. Fig. 5I, from the MD, suggests the C-terminus-C-terminus distance is 10 nm while it is <1 nm for the N-termini. The actual FRET signals are not very different at all, certainly not comparable to what might be predicted on the basis of the differing separations.

Answer: We copy here an experimental figure from the landmark paper of Stryer “Energy transfer: a spectroscopic ruler”. PNAS 1967

FIG. 4.—Efficiency of energy transfer as a function of distance in dansyl-(L-prolyl) $_n$ - α -naphthyl, $n = 1$ to 12. The α -naphthyl and dansyl groups were separated by defined distances ranging from 12 to 46 Å. The energy transfer is 50% efficient at 34.6 Å. The solid line corresponds to an r^{-6} distance dependence.

The drop in FRET as observed in this chemically well-defined system occurs within a range of 2 to 5 nm, a range well adapted for the distances we want to assess in our protein constructs (indeed a function close to $r^6/(R_0^6 + r^6)$). However, we apologize for not having specified that the acceptor-donor probes on the G324C construction are not located on the C-terminus separated by 10 nm but on the putative H2 helix and separated by 7 nm.

Moreover, it seems that the process of monomer exchange between dimers is long. In the case of N-terminus constructions, the plateau is reached after 500-600 minutes. The value of FRET (AU) is 1.9 while it is 1.3

(AU) for the C-ter construct (i.e. a 3-fold difference). The same difference in FRET is observed in a more directed experimental approach, that does not rely on monomer exchange kinetics and that is now presented in a new panel in the same figure, further strengthening the reliability of the observed values.

We have modified figure 5, and changed the text “no FRET” by “low FRET”, so that the Gly324 will appear on the dimer model in 5H, and we have inserted the distance curve between the probes in position 324 during the energy minimization Fig. 5I.

In the discussion we modified the sentence: Biochemical analysis indicates that the region between the PH domain and FFAT motif is dimeric, alpha-helical and has its N-termini very close together, while its C-termini are distant far away.

We kept the sentence that expresses that we are aware that we are far from having an exact model for the central core OSBP, but also that whatever the exact model, the T shape has consequence on the functioning of OSBP:

“Regardless the exact atomic structure, which will require further studies, the consequence of this organization is that the tethering and lipid transfer moieties of each monomer are separated by a large distance”

Reviewer #1 (Remarks to the Author):

The authors responded satisfactorily to most of my comments:

Q1: OK

Q2: Regarding the direct interaction between OSBP and VAP-A they have pointed out previous experiments that supports this, in addition to successfully performing flotation assays.

Q3: I agree there is a strong suggestion, supported by the new experiments with the MSP-only construct that the MSP does indeed localise to the tip of the molecule. I still disagree with its direct assignment to 'dark dots', as this suggests that based on intensity one is able to discriminate between different domains of a molecule, which is not the case. I suggest rephrasing to: "Previous structural studies and our model of VAP-A suggest that the dark distal densities are likely to correspond to the MSP domain", or something like this.

Q4: OK

Q5: This is a good addition, and makes the analysis stronger.

Q6: OK

Q7,8: I am satisfied with the discussion about thresholding. We all agree that at this resolution this is bound to be partly subjective. I think the manuscript would improve if the authors included some discussion on how threshold was determined in the methods.

Q9: "The reviewer is expressing two points of concern: first that the reported resolution does not correspond to the visual appearance of the map; second that the map might contain some degree of template bias."

I have to disagree with the concept that these are different concerns: they are both aspects of overfitting and must be considered together.

I am not concerned about the absolute number for the resolution, but by the fact that an overestimation is a sign of a problem in the processing.

If the authors can exclude overfitting because they have carried out their analysis following a strict gold standard then this must be specified in the methods: at what point the dataset was split, and what was the low-pass filter applied to the initial templates.

The theory cited from Chen et al. does indeed mean that the resolution is mask independent when using phase-randomisation, in the sense that the mask itself does not affect the FSC. However, the resolution is indirectly dependent on the mask if the masks used exclude different amounts of poorly resolved non-solvent region, for example parts of the membrane or neighbouring molecules. So it is not a surprise at all that different masks give different resolutions in this case. It would be worrying if this was not the case.

Q10: I appreciate that positions of aligned subtomograms were plotted to exclude clashes. It is hard to see from figure Q10.2: do the relative positions of VAP-A and N-PH-FFAT indicate they form

contacts (e.g. pairs)? They look rather randomly scattered but the picture is not very clear. I wonder whether it would be possible to analyse their relative distribution: for instance, if the positions of adjacent N-PH-FFAT molecules is plotted for each VAP-A, do they show preferential relative positions? If not, why?

Q11: I still think the model is quite weak. I think the authors should discuss in the last paragraph of their results that they present what they consider the most likely model, but cannot exclude that other models are possible.

Reviewer #2 (Remarks to the Author):

The authors have addressed all concerns by this reviewer. I recommend publication.

Reviewer #3 (Remarks to the Author):

The manuscript by de la Mora and colleagues has undergone some significant improvements and clarifications. Particularly, the addition around line 209 comparing the proportions of MCS structures obtained in the presence of tubes containing N-PH-FFAT with those obtained in the presence of full length OSBP and the reworking of Fig. 5D and E, which is particularly helpful.

The detailed and careful work done in the rebuttal has clarified the work and some of the questions raised. Indeed, my only recommendation at this time is to add some of the experiments from the rebuttal to the supplementary information as they give useful insight to the reader and strengthen the foundation of the manuscript. These would be the gel shown in response to Q1 (in response to queries both by reviewer #1 and myself), where the dimer disappears on boiling and the DLS data. Actually, also the orientation data. Maybe space can be found for these in the SI.

Lines 192-196: rework – the intended meaning of the authors is entirely unclear.

Lines 309-314: several mis-referencings of the relevant figures

Lines 397-399: rework

Overall, this remains an exciting study and I congratulate the authors on an important and technically demanding piece of work.

Revision 2.

We would like to thank the reviewers for their insightful comments. Please find below our point-by-point response. We identify each question/comment and our response by a number from Q1 to Q15. The revision includes the following changes:

Reviewer 1: he/she agreed with your previous answers of Question Q1/Q2/Q4/Q5/6. We now modified the text to answer to question Q3/Q9/Q11. We also computed data to answer Q10.

Reviewer 2: no question

Reviewer 3: We modified the text to answer to questions Q12/Q13/Q14/Q15. We modified the figure S1 with additional results as suggested in Q11. We added these new results in the source data file.

Looking forward to seeing your feedback,

Sincerely,

Daniel Levy

REVIEWER COMMENTS

Reviewer #1 (Remarks to the Author):

The authors responded satisfactorily to most of my comments:

Thank you

Q1: OK

Q2: Regarding the direct interaction between OSBP and VAP-A they have pointed out previous experiments that supports this, in addition to successfully performing flotation assays.

Q3: I agree there is a strong suggestion, supported by the new experiments with the MSP-only construct that the MSP does indeed localise to the tip of the molecule. I still disagree with its direct assignment to 'dark dots', as this suggests that based on intensity one is able to discriminate between different domains of a molecule, which is not the case. I suggest rephrasing to: "Previous structural studies and our model of VAP-A suggest that the dark distal densities are likely to correspond to the MSP domain", or something like this.

We have followed your suggestion and have written:

Lines 139-140

Previous structural studies^{13,26,31} and our model of VAP-A (see below Figure 4) suggest that the dark distal densities are likely to correspond to the MSP domain.

Q4: OK

Q5: This is a good addition, and makes the analysis stronger.

Q6: OK

Q7,8: I am satisfied with the discussion about thresholding. We all agree that at this resolution this is bound to be partly subjective. I think the manuscript would improve if the authors included some discussion on how threshold was determined in the methods.

*We have now rewritten in the **Visualization** paragraph as:*

Lines 676-685. Rendering of 3D density maps is performed with UCSF Chimera⁶⁹. The threshold of intensity was defined as it corresponded to a point where the intensity distribution in the map showed a change of derivative. Rigid-body docking of the crystallographic structure of the dimer of VAP-A MSP (PDB ID code 1Z90) and of the computed (195-335)-OSBP dimer into the cryo-EM density maps was done with Chimera. The dimer of MSP was selected over the monomer based on the correlation calculated by Chimera, 0.93 for the dimer vs 0.89 for the monomer, after fitting the model into the map by simulating a map/calculating a simulated map at 20 Å. The lower correlation was related to the lower mass of the monomer compared to that of the calculated map. A tetramer was also fitted into the map but its volume surpassed that of the calculated map. The estimated correlation was 0.80 in this case.

Q9: “The reviewer is expressing two points of concern: first that the reported resolution does not correspond to the visual appearance of the map; second that the map might contain some degree of template bias.”

I have to disagree with the concept that these are different concerns: they are both aspects of overfitting and must be considered together.

I am not concerned about the absolute number for the resolution, but by the fact that an overestimation is a sign of a problem in the processing.

If the authors can exclude overfitting because they have carried out their analysis following a strict gold standard then this must be specified in the methods: at what point the dataset was split, and what was the low-pass filter applied to the initial templates.

We now wrote lines 644-648:

The overall resolutions for VAP-A and of NPH-FFAT as determined by splitting and analyze the data set at step 5 of the work-flow from the gold standard Fourier Shell Correlation (FSC) processing **using FSC0.134 was 19.6 Å and 9.8 Å, respectively (Figure S4). For NPH-FFAT, a further inspection of the local resolution computed by the program blocres (BSoft)⁵⁹ pointed at a distribution of resolutions in the range of 9-18Å**

A low pass-filter of 40 Å has been applied as it was written in the work-flow depicted Figure S4, at step 4, step 5 and step 6.

The theory cited from Chen et al. does indeed mean that the resolution is mask independent when using phase-randomisation, in the sense that the mask itself does not affect the FSC. However, the resolution is indirectly dependent on the mask if the masks used exclude different amounts of poorly resolved non-solvent region, for example parts of the membrane or neighboring molecules. So it is not a surprise at all that different masks give different resolutions in this case. It would be worrying if this was not the case.

Q10: I appreciate that positions of aligned subtomograms were plotted to exclude clashes. It is hard to see from figure Q10.2: do the relative positions of VAP-A and N-PH-FFAT indicate they form contacts

(e.g. pairs)? They look rather randomly scattered but the picture is not very clear. I wonder whether it would be possible to analyse their relative distribution: for instance, if the positions of adjacent N-PH-FFAT molecules is plotted for each VAP-A, do they show preferential relative positions? If not, why?

The image below represents a 2d histogram, the color of each pixel scales as the number of couples of particles (one VAP-A and one NPHFFAT) that are found at the given distance (in the x axis in Å) and the given deviation from co-linearity (in the y axis in degrees). For many particles, there is a strong correlation between mutual position and mutual orientation for the two particles in the pair while for others orientation and distance from neighboring proteins of the other species is more random. This is consistent with the variability found in the MCSs that limits our resolution. As presented in the discussion, some reasons could be evoked: - i) the presence of a long disordered region encompassing the FFAT motif (aa 325-408) at the C-terminus of N-PH-FFAT that i.e. 31 aa before the FFAT motif, and that might extend up to 10. **We previously wrote lines 413-416:** “As a result, the orientation and position of the T domain relative to VAP-A might vary. This is what we observed when analyzing a **single** 3D class **as** defined by the fixed intermembrane distance: the VAP-N-PH-FFAT complexes were not at fixed positions thus limiting the resolution of the whole complex”, ii) the possibility that the two disordered regions of N-PH-FFAT dimer interact with one or two VAP dimers simultaneously. **We have now added line 416:** In addition, this opens the possibility that the two disordered regions of N-PH-FFAT dimer interact with one or two VAP dimers simultaneously.

2D plot of relative orientation and distances between VAP-A and N-PH-FFAT in MCSs. The number of couples of particles is depicted as color code.

Q11: I still think the model is quite weak. I think the authors should discuss in the last paragraph of their results that they present what they consider the most likely model, but cannot exclude that other models are possible.

We have now written:

Line 325-327

Thus, this model seemed as the most likely but since the resolution was not enough to assign secondary structures, we cannot exclude that other models are possible. Finally, the stem of the T that connects to the membrane remaining to be assigned.

We previously wrote in the discussion and keep it in the revised version the possibility that other models cannot be excluded:

Lines 408-410: Regardless the exact atomic structure, which will require further studies, the consequence of this organization is that the tethering and lipid transfer moieties of each monomer are separated by a large distance, akin to the jib of a tower crane.

Reviewer #2 (Remarks to the Author):

The authors have addressed all concerns by this reviewer. I recommend publication.

Thank you

Reviewer #3 (Remarks to the Author):

The manuscript by de la Mora and colleagues has undergone some significant improvements and clarifications. Particularly, the addition around line 209 comparing the proportions of MCS structures obtained in the presence of tubes containing N-PH-FFAT with those obtained in the presence of full length OSBP and the reworking of Fig. 5D and E, which is particularly helpful.

Q12: The detailed and careful work done in the rebuttal has clarified the work and some of the questions raised. Indeed, my only recommendation at this time is to add some of the experiments from the rebuttal to the supplementary information as they give useful insight to the reader and strengthen the foundation of the manuscript. These would be the gel shown in response to Q1 (in response to queries both by reviewer #1 and myself), where the dimer disappears on boiling and the DLS data. Actually, also the orientation data. Maybe space can be found for these in the SI.

We have now added in figure S1 the SDS PAGE gel in the presence of reducing agents and the gels of trypsin digestion for the determination of the orientation. Accordingly, we have written in the text

Line 94: In SDS PAGE gel and in the presence of reducing agents, VAP-A preparation showed two bands at ~ 55kDa and ~ 25 kDa, while a single band was found when the samples were heated revealing dimers of VAP-A in the preparation (Figure S1A).

Line 196: We first reconstituted VAP-A at LPR 70 mol/mol to obtain proteoliposomes with various VAP-A densities. Then, they were mixed either with N-PH-FFAT or OSBP bound to PI4P Galcer tubes. The resulting MCSs were analyzed by cryo-tomography. **As shown by trypsin digestion, VAP-A was symmetrically oriented in proteoliposomes (Figure S1G)** but only VAP-A pointing outward from the vesicles was engaged in MCS formation.

Q13: Lines 196-200: rework – the intended meaning of the authors is entirely unclear. *We have now written*

We first reconstituted VAP-A at LPR 70 mol/mol to obtain proteoliposomes with various VAP-A densities. Then, they were mixed either with N-PH-FFAT or OSBP bound to PI4P Galcer tubes. The resulting MCSs were analyzed by cryo-tomography. As shown by trypsin digestion, VAP-A was symmetrically oriented in proteoliposomes (Figure S1G) but only VAP-A pointing outward from the vesicles was engaged in MCS formation.

Q14: Lines 309-314: several mis-referencings of the relevant figures

Corrected. Thank you.

Q15: Lines 397-399: rework.

We have now written

Lines 402-405: The tethering region of OSBP is organized as a T-shaped with a 14 nm elongated domain parallel to the membrane and a short ≈ 3 nm stem. The resolution is not enough to assign a specific domain of OSBP but since it connects to the membrane, it probably contains part of the two PH domains.

Overall, this remains an exciting study and I congratulate the authors on an important and technically demanding piece of work.

Thank you

Reviewer #1 (Remarks to the Author):

I am happy with the further revision and I recommend publication.

Reviewer #3 (Remarks to the Author):

I congratulate the authors on an exciting and important study and look forward to the next chapters. The small issues I have suggested previously have been addressed in full.